# Redox-active electrolyte-based printed ionologic devices

Hanfeng Zhou[1], Przemyslaw Galek [1]✉, Tianle Zheng [2], Panlong Li[1], Xiongjun Zhou[3], Congcong Liu[4], Jonas Kunigkeit[5], Katherina Haase[6], Yuxi Li[1], Jiang Qu [4], Ahmed Bahrawy[1], Peixun Xiong [1], Julia Grothe[1], Daria Mikhailova[4], Stefan C. B. Mannsfeld [6], Eike Brunner[5] & Stefan Kaskel [1,7]✉

Ionic devices, such as electrochemical capacitor diodes (CAPodes) and gate-controlled CAPodes with transistor-like gating characteristics (G-CAPodes), offer a novel approach to energy-efficient and nature-inspired logic computing. Their miniaturization and integration render them ideal for ion-transistor circuits, enabling the regulation and signaling of ions and biomolecules. Here, we report an asymmetric system to achieve a potential-driven ion pump for CAPode based on a redox-active Keggin-type electrolyte. This unidirectional capacity is achieved through asymmetric polarization between a plane metal and a porous carbon electrode, enabling selective redox reactions on the metal surface. The nanoporous carbon effectively balances the charge on the redox electrode, while redox couples control the working voltage range. Printed ionologic devices are demonstrated for logic gates, and an integrated NAND (NOT-AND) circuit was constructed using two CAPodes and one G-CAPode. This work proposes a concept for switchable iontronic devices, providing a deeper understanding and applicability of these devices.

The precise handling and propagation of ionic/molecular signals play a pivotal role across life science disciplines[1]. This technological imperative has driven the development of ion-transistor networks capable of orchestrating complex biochemical patterns with spatio-temporal precision. While their operational speeds lag behind electronic systems due to fundamental ion transport constraints, these platforms uniquely decode chemical information[2]. This capability for molecular-level recognition allows for targeted biological modulation, particularly for: (i) neural interface applications (seconds to hours)[3], (ii) dynamic cellular signaling like $Ca^{2+}$ waves (μs-day intervals)[4], and (iii) plant morphogenesis regulation (min-day periods)[5].

To date, several ionologic devices with various functioning processes and architectures have been developed. Among them, electrochemical capacitors (ECs)-based CAPodes and G-CAPodes with supercapacitance, low threshold voltage, minimal energy consumption, and high frequency-dependent impedance characteristics (Tab. S3–S4), are the most promising contenders. These systems offer high rectification, long lifespan, and superior durability for applications like neural recording analog front-end, sensor read-outs, and bio-amplifiers, where precise control over frequency-dependent resistance is crucial[6–8]. These capacitive logic elements represent a form of ionic circuit, aiming to utilize ions as charge carriers in solutions for signal processing. In electrolytic solutions, fluidic guides and ion-selective media have been arranged to realize ionic diodes and ionic transistors as fundamental circuit components[9–15]. Meanwhile, various CAPode and G-CAPodes designs have been proposed based on mechanisms such as electric double-layer (EDL) formation, surface redox, or intercalation[8,16–18]. Particularly, surface-redox CAPodes exhibit

[1]Inorganic Chemistry I, Technische Universität Dresden, Dresden, Germany. [2]Department of Chemistry, Shanghai University, Shanghai, China. [3]Mechanical and Electrical Engineering, Kunming University of Science and Technology, Kunming, China. [4]Leibniz Institute for Solid State and Materials Research, Dresden, Germany. [5]Bioanalytical Chemistry, Technische Universität Dresden, Dresden, Germany. [6]Center for Advancing Electronics Dresden, Technische Universität Dresden, Dresden, Germany. [7]Fraunhofer Institute for Material and Beam Technology, Dresden, Germany. ✉e-mail: przemyslaw.galek@tu-dresden.de; stefan.kaskel@tu-dresden.de

improved rectification capability owing to their synthesized working electrodes with intrinsic nanostructure. However, not only is the cost of the synthesis process high, but the working voltage window is also limited by electrode stability. Therefore, the rational adjustment of redox potential relationship for electrolyte ions and the difference in polarization between electrodes unlocks the full performance of CAPode/G-CAPode with a higher adjustable voltage range.

Furthermore, the miniaturization for integration purposes is the ultimate goal of ion-based computing research for CAPodes and G-CAPodes. Recent efforts have significantly advanced the miniaturization of 2D in-plane interdigital ECs composed of carbon materials[19–21]. Printing serves as the foundation for the miniaturization of ionologic devices based on carbon materials[22,23]. More notably, advanced printing techniques have been utilized to manufacture two different ionologic devices consisting of carbon materials. A G-CAPode was realized as a three-terminal functional architecture based on a protonic gel electrolyte[23], and a bioinspired implantable G-CAPode was printed using a piezoelectric printing technique[22]. These printed carbon-based ionologic systems provide a viable foundation for exploring the possibilities of miniaturizing additional carbon-based ionologic devices.

Accordingly, we present herein an asymmetric cell system designed to elicit a selective capacitive response from redox ions. While asymmetric capacitors with one redox-active solid electrode have been reported before[17], the novelty of the approach presented here is the electrolyte ion itself, which is redox active. In this design, the faradaic process occurs only on one of the electrodes (planar Ti), while the porous counter electrode (CE) acts purely as a capacitive Keggin-ion electrosorption material. The significant variation in specific surface area between the two electrodes is responsible for achieving asymmetric polarization profiles for the metal and carbon electrodes. This versatile system is compatible with various redox molecules, and the working voltage range can be regulated by selecting different redox couples. CAPode, and G-CAPode were fabricated using screen-printing techniques. Their excellent rectification capability and electrochemical performances enable high efficiency in AND, OR, NOT, NAND logic gates, as well as their integration, demonstrating significant potential in ion/electron coupling logic operations.

## Results

### System construction

Materials and electrochemical measurements are detailed in Sections 1 and 2 in the supplementary information (SI), respectively. The proposed CAPode consists of a plane Ti metal electrode and a porous carbon-based electrode with high specific surface area (SSA).

Activated nanoporous carbon ($C_x$; where $x$ is the average size of pores in nm) with a high SSA is used as an actively involved material in charge storage at a carbon-based electrode. A redox-active aqueous medium is used as an electrolyte. The proposed asymmetric CAPode cell can be symbolized as: (Ti | redox electrolyte | carbon@metal). The asymmetry in this system ensures a unique ion-selective, intriguing redox reaction, which can only occur on the metal electrode. The rectification capability of such CAPode is expected to be enormously improved owing to selective redox reactions. To validate this concept, Ti and $C_{1.5}$ are used as electrode materials due to their high availability, safety, and non-toxicity to construct the (Ti | electrolyte | $C_{1.5}$) CAPode. Details on assembling the system can be found in Section 3 in SI. $N_2$ physisorption isotherm for $C_{1.5}$ and its textural properties are presented in Fig. S1 and Tab. S1, respectively (in Section 4 in SI). Furthermore, the existence of $TiO_2$ on the surface of Ti mesh was confirmed by the Raman spectrum (Fig. S2 in Section 5 in SI). As an electrolyte 1 M phosphotungstic acid ($H_3PW_{12}O_{40}$; PWA) with Keggin structure in water is ideal due to its unique reversible redox characteristics, crucial for achieving high rectification ratio ($RR$) and unidirectional charge storage in CAPode. Additionally, the high proton conductivity and electrochemical stability of PWA enhance the overall performance and reliability of the system[24–26]. The electrolyte anion has tetrahedral symmetry, with twelve tungsten atoms linked by oxygen and a phosphorus atom at its center.

### Principle of operation

When the CAPode (in 2-electrode setup; Fig. S3 in Section 6 in SI) is working under "open" (chargeable) polarization conditions (0 ↔ −1 V; left scheme in Fig. 1a), the Ti electrode is negatively (−) polarized and $C_{1.5}$ electrode positively (+). The $PW_{12}O_{40}^{3-}$ anions are potentially electrostatically adsorbed on the Ti surface, and

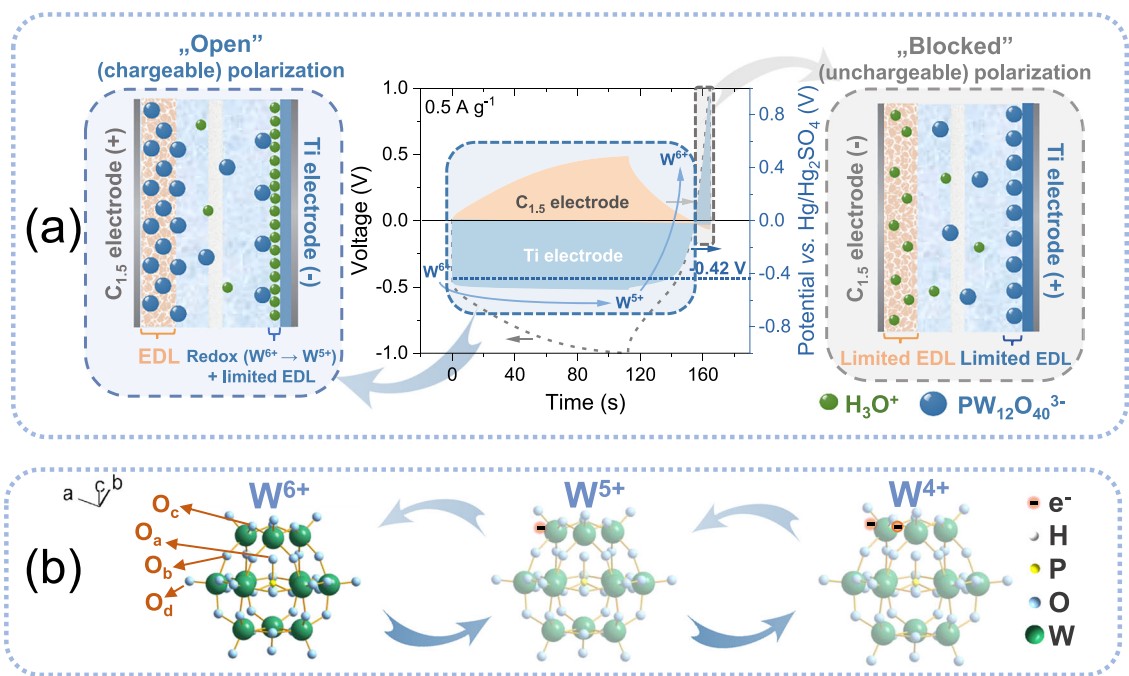

**Fig. 1 | Principle of operation for CAPode. a** Galvanostatic charge/discharge (GCD) curve for repolarized CAPode (Ti | 1 M $H_3PW_{12}O_{40}$ | $C_{1.5}$) and schematic illustration showing the charge storage mechanism. **b** Redox reactions on the Ti electrode for the proposed CAPode.

their oxidation state VI ($W^{VI}$) is partially reduced to $W^V$. This redox process, described in Eq. 1 and illustrated in Fig. 1b, highlights the reversible electron transfer within the PWA framework, showcasing its structural stability and electron-storage[24–27].

$$PW_{12}^{VI}O_{40}^{3-} + 1e^- \rightleftharpoons PW_{11}^{VI}W^VO_{40}^{4-} \qquad (1)$$

This reaction occurs because under "open" polarization Ti electrode can reach the initial redox potential −0.42 V *vs.* reference electrode (RE; $Hg/Hg_2SO_4$ with saturated $K_2SO_4$) in the GCD curve registered for the repolarized CAPode system (middle plot in Fig. 1a). Meanwhile, the $C_{1.5}$ electrode is (+) polarized, and a large number of $PW_{12}O_{40}^{3-}$ anions balance the charge on the Ti(−) electrode through EDL formation at the high SSA of the $C_{1.5}$ electrode. Therefore, the full cell can effectively be charged (has high capacity).

However, when the CAPode is polarized under "blocked" (non-chargeable) polarization (the right scheme in Fig. 1b), the Ti electrode becomes (+) polarized. Since the $C_{1.5}(−)$ electrode has a high SSA (compared to Ti mesh (9.97 $m^2$ with mass loading 5 mg)), it is only slightly polarized and cannot reach the low redox potential required for faradaic $W^{6+}/W^{5+}$ reduction. However, the applied 1 V voltage is readily reached with little current, as the Ti electrode with low SSA cannot capacitively store anions and is rapidly overpolarized. Under these conditions, the CAPode capacitance is limited to EDL formed on a very low Ti electrode SA (<0.6 $cm^2$), and that is why the system is not effectively charged (blocked). Additionally, DFT calculations (experimental details can be found in Section 7 in SI) show the relatively low LUMO of PWA (−6.40 eV, in Figs. S4a and S5) indicates that the molecule is easily reduced.

Overall, due to the specific polarization of the Ti and carbon electrode when the system is repolarized, the constructed CAPode can effectively store the charge as a result of a redox reaction only when the Ti electrode is negatively polarized.

## Electrochemical performance

For further electrochemical investigation, a redox-active PWA electrolyte (1 M; pH = 0.54; Fig. S6 in Section 8 in SI) was chosen. As shown in Fig. S7a (in Section 9 in SI), obvious redox peaks appear in the cyclic voltammetry (CV) curves at various scan rates for the Ti electrode (in a 3-electrode setup), reflecting the redox-dominated charge storage mechanism in the PWA electrolyte. More detailed information about the charge storage mechanism of the proposed CAPode can be found later in Section 9 in SI (Figs. S7b–f and S8).

Rectification ratio I ($RR_I$; Eq. S6) and rectification ratio II ($RR_{II}$; Eq. S7, Section 10 in SI), were used to evaluate the system' rectification performance. Rectification coefficients are fundamental parameters that define the performance of CAPode systems. The $RR_I$ value is calculated as the ratio of currents at a specified voltage, typically the highest applied, under "open" and "blocked" polarization, respectively. $RR_{II}$ represents the ratio of capacitance under "open" conditions to the total capacitance, combining both "open" and "blocked" conditions. To confirm the rectification performance, we calculated $RR_I$ and $RR_{II}$ for the entire range of applied scan rates (Fig. S9).

To further investigate the ion storage mechanism (Fig. 2), we constructed a 2-electrode setup with RE (Fig. S3) to simultaneously monitor the potential distribution between the working (WE) and CE and to control only one electrode.

Figure 2a shows the potential distribution between Ti(−) and $C_{1.5}(+)$ electrode in the CAPode system (as GCD profiles registered at 0.5 A $g^{-1}$) under "open" polarization (from open-circuit voltage (OCV) −0.16 V; see detailed explanation of this approach in Section 9 in SI). The $C_{1.5}(+)$ electrode stores charge through EDL formation on its surface, which can be observed as a triangular shape of the charge/discharge profile (red profile). The Ti(−) electrode (blue curve) exhibits

battery-like behavior with a potential plateau, where $W^{6+}$ ions undergo previously described redox reactions during the charge and discharge process. CV provides a clearer picture of the different types of electrochemical processes that occur on the electrodes. The different charge storage mechanism on both electrodes is confirmed by CV curves registered for the $C_{1.5}(+)$ (at 2.6 mV $s^{-1}$) and Ti(−) (at 4.8 mV $s^{-1}$) electrode separately (the scan rates were calculated based on galvanostatic discharge profiles) presented in Fig. 2b. On the CV curve registered for the Ti(−) electrode distinct peaks indicating the faradaic processes are visible, while the CV curve shape for the $C_{1.5}(+)$ electrode is close to the rectangular indicating EDL formation on its surface.

To facilitate discussion of the data and comparison, symmetric cells ($C_{1.5}$ | 1 M PWA | $C_{1.5}$ and Ti | 1 M PWA | Ti) were also evaluated (Fig. 2c) in 2-electrode setups (Fig. S3). The symmetric $C_{1.5}$-based cell reveals a rectangular-shaped CV curve (red line) under both opposite polarization directions, as expected for typical EDL behavior. No matter in which direction the electrodes are polarized, the charge is effectively stored at the high SSA of the porous carbon electrodes. However, slight redox peaks are also visible, particularly when the system is subjected to a slower scan rate (5 mV $s^{-1}$; Fig. S10a). In the symmetric $C_{1.5}$-based system, the potential of both electrodes can reach the redox potential (−0.45 V *vs.* RE; Fig. S10b), which means that the redox reaction can occur on the surface. However, it should be mentioned here that the overall electrochemical response of the system is dominated by the EDL capacitance. The total contribution of the redox response is small and was registered only on one of the electrodes as a peak in the blue CV curve in Fig. S10c after surpassing the potential of the redox potential. On the other hand, the Ti-based symmetric cell exhibits an exclusive negligible current response (Fig. 2c). The CV curves have a quadrangular shape with low current response, which is typical for the capacitive processes that occur during the EDL formation on a limited SSA (Fig. S11). None of the electrodes reaches the redox potential as it is presented in GCD profiles in Fig. 2d (0.06 mA $cm^{-2}$ was applied to obtain a charge/discharge time similar to the asymmetric cell as in Fig. 2a). Moreover, theoretically, there is no possibility that the planar Ti(+) electrode can balance the charge that could be generated on the redox Ti(−) electrode, unlike the asymmetric system in Fig. 2a. Figure S12 shows the CV curves corresponding to different symmetric cell configurations ($C_{1.5}$, steel, and Ti) at a scan rate of 2–100 mV $s^{-1}$. None of these systems exhibits behavior comparable to the proposed asymmetric CAPode system.

To confirm the anion accessibility and adsorption mechanism of $C_{1.5}$, the CAPode (Ti | 1 M PWA (deuterated $H_2O$) (d-$H_2O$) | $C_{1.5}$) was monitored by $^{31}P$ magic angle spinning nuclear magnetic resonance (MAS NMR) spectroscopy (details of this experiment can be found in Section 13 in SI). Notably, a $^{31}P$ MAS NMR signal is present at a chemical shift of −19.5 ppm, i.e., lower than the signal for the pure electrolyte at −15.2 ppm (Figs. 2e and S13. This shift is induced by the well-known so-called ring current effect[28–30]. The shifted signal denoted as 3 in Fig. 2e represents ~50% of the signal intensity, i.e., its intensity is approximately equal to the sum of species 1 and 2, i.e., free bulk and bulk species in contact with the outer particle surface. This observation indicates the presence of a considerable number of electrolyte ions entering the larger pores present in the $C_{1.5}$ electrode (diameter of PWA = 1.5 nm, pore volume larger than 1.5 nm = 0.63 $cm^3$ $g^{-1}$; 47% to total pore volume). That means, the NMR experiments reveal that ion adsorption in the pore system of $C_{1.5}$ electrode happened here in the absence of an external voltage, and the anions can access the pores. The CAPode can thus, in principle, store charges if the bulky anions are attracted by the positively polarized $C_{1.5}(+)$ electrode.

We evaluated the performance of various (Ti | 1 M PWA ($H_2O$) | $C_x$) CAPodes based on different pore sizes of the $C_x$ electrode (Fig. 2f). Interestingly, the current response (especially under "open" polarization) of the system with $C_{1.5}$ electrode is much larger than with $C_{0.70}$ and $C_{0.55}$ at the same scan rate (20 mV $s^{-1}$). This may result from the

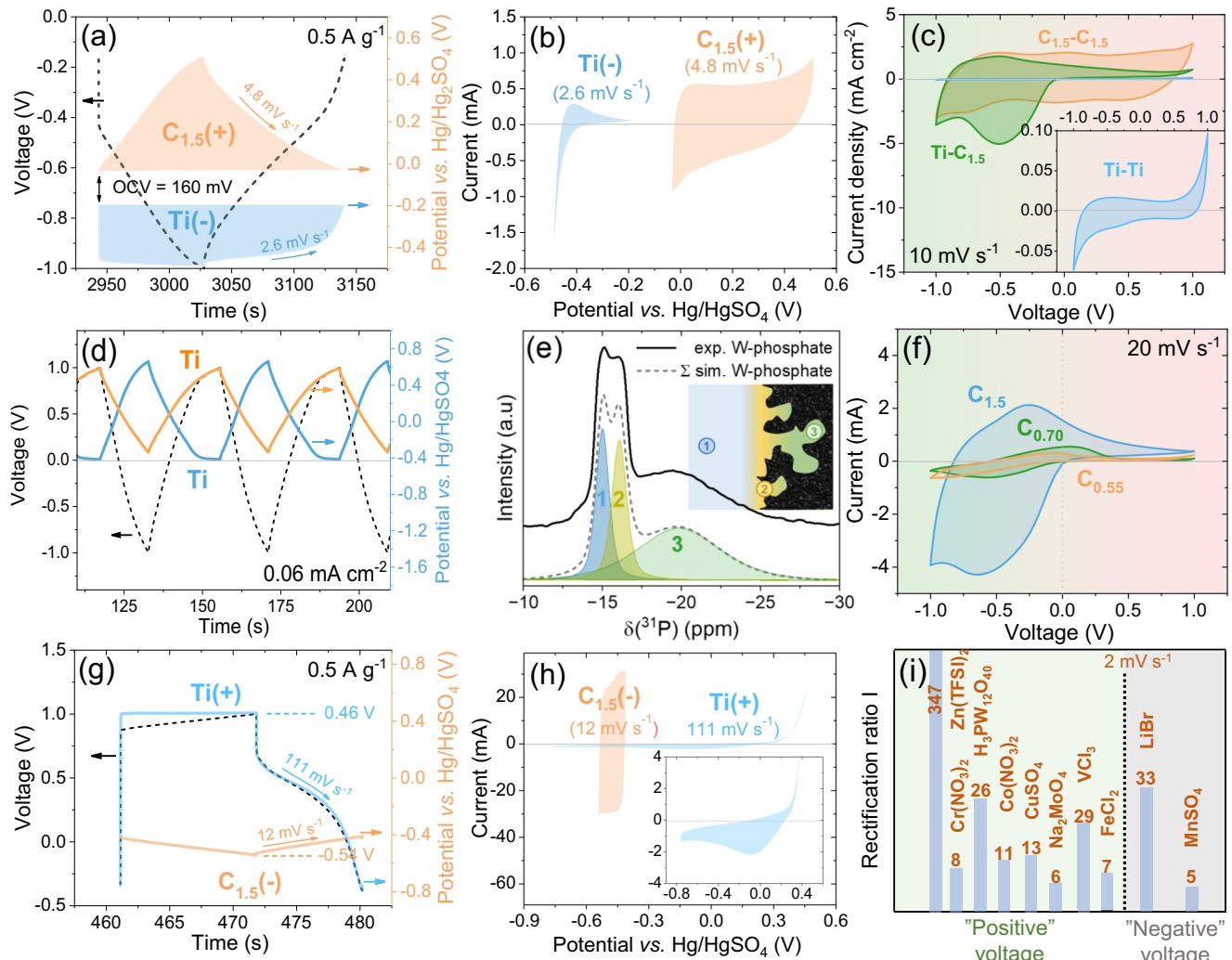

**Fig. 2 | Electrochemical performance of CAPodes. a** GCD profiles and **b** CV curves for the $C_{1.5}(+)$ and Ti(−) electrode under "open" polarization. **c** Comparison of CV curves for (Ti | 1 M PWA | $C_{1.5}$), (Ti | 1 M PWA | Ti), and ($C_{1.5}$ | 1 M PWA | $C_{1.5}$). **d** The GCD profiles for (Ti | 1 M PWA | Ti). **e** $^{31}P$ MAS NMR spectra of $C_{1.5}$ loaded with 1 M PWA in

d-$H_2O$. **f** Comparison of CV curves of carbons with various SSA and pore size distribution. **g** The GCD profiles and **h** CV curves for the $C_{1.5}(−)$ and Ti(+) electrode under "open" polarization by using 1 M LiBr in $H_2SO_4$ as the electrolyte. **i** $RR_I$ comparison for Ti-$C_{1.5}$ system with different redox-active salts as electrolytes.

fact that $C_{0.70}$ and $C_{0.55}$ materials have much lower pore sizes and SSA (1616 $m^2 g^{-1}$ for $C_{0.70}$ and $C_{0.55}$ and 500 $m^2 g^{-1}$; Tab. S1) than $C_{1.5}$ (2492 $m^2 g^{-1}$; Tab. S1). Therefore, adjusting the SSA of carbon electrodes can also regulate the EDL, which can balance charges from the faradaic redox process of $W^{6+}/W^{5+}$, optimizing the charge storage. In particular, the presence of both micropores and mesopores in $C_{1.5}$ may facilitate proton conduction and enhance redox reactions of PWA. Furthermore, a large amount of mesopores (0.24 $cm^3 g^{-1}$) in $C_{1.5}$ facilitates rapid transport of ions, which is crucial for maintaining high EDL capacitance to balance the charges from the redox reaction[31]. Furthermore, the redox peaks of $W^{6+}/W^{5+}$ are shifted (Fig. 2f), which indicates that the SSA and pore size of carbon can affect the polarization of the carbon electrode.

Figure 2g shows the GCD profiles for the asymmetric cell LiBr at 0.5 A $g^{-1}$. The Ti(+) electrode (blue line) stores charge through a faradaic process (plateau) where $Br^-$ anions undergo redox processes during charge and discharge, whereas the $C_{1.5}(−)$ electrode displays a purely capacitive. The curve for the Ti(+) electrode (Fig. 2h) demonstrates quasi-reversible single-electron transfer with -0.5 V peak, probably due to the slow mass transfer in this viscous medium, while the $C_{1.5}(−)$ electrode displays a quadrangular shape, typical of capacitive formation of the EDL. Several redox-active electrolytes were tested

in our CAPode (Figs. 2i and S14) for a comparison of their $RR$. Fig. S15 shows a decrease in current response with CV cycles in the $Zn(TFSI)_2$ system, likely due to Zn plating/stripping ($Zn^{2+}/Zn^0$), forming dendrites and by-products that degrade cycling stability and CAPode performance, and the increased peak separation in CV curves at 100 mV $s^{-1}$ in the LiBr system, which indicates incomplete reversibility of the redox reaction. The PWA was selected as the optimized electrolyte for the CAPode application because it provides high $RR$ and is characterized by low-cost, high proton conductivity, redox reversibility, and electrochemical stability. These PWA properties enhance the overall performance and reliability of the system[24–26]. Such a system can only present a high current response under "open" polarization. A more detailed discussion of the advantages of PWA can be found in Section 13 in SI. In contrast, the two CAPode systems based on $Zn(TFSI)_2$ and LiBr, despite their high $RR$, suffer from significant instability after several cycles of repolarization (Fig. S15).

The rectification mechanism was investigated using a customized two-electrode cell through ex-situ and in situ measurements (Figs. S16–S20). In situ Raman spectroscopy revealed a new broad band at 92 $cm^{-1}$ at −0.8 V (Fig. S16a–c), while in situ X-ray absorption spectroscopic studies (XAS) of the W $L3$-edge showed a lower XANES energy for PWA under bias (10199.97 eV) compared to $WO_3$

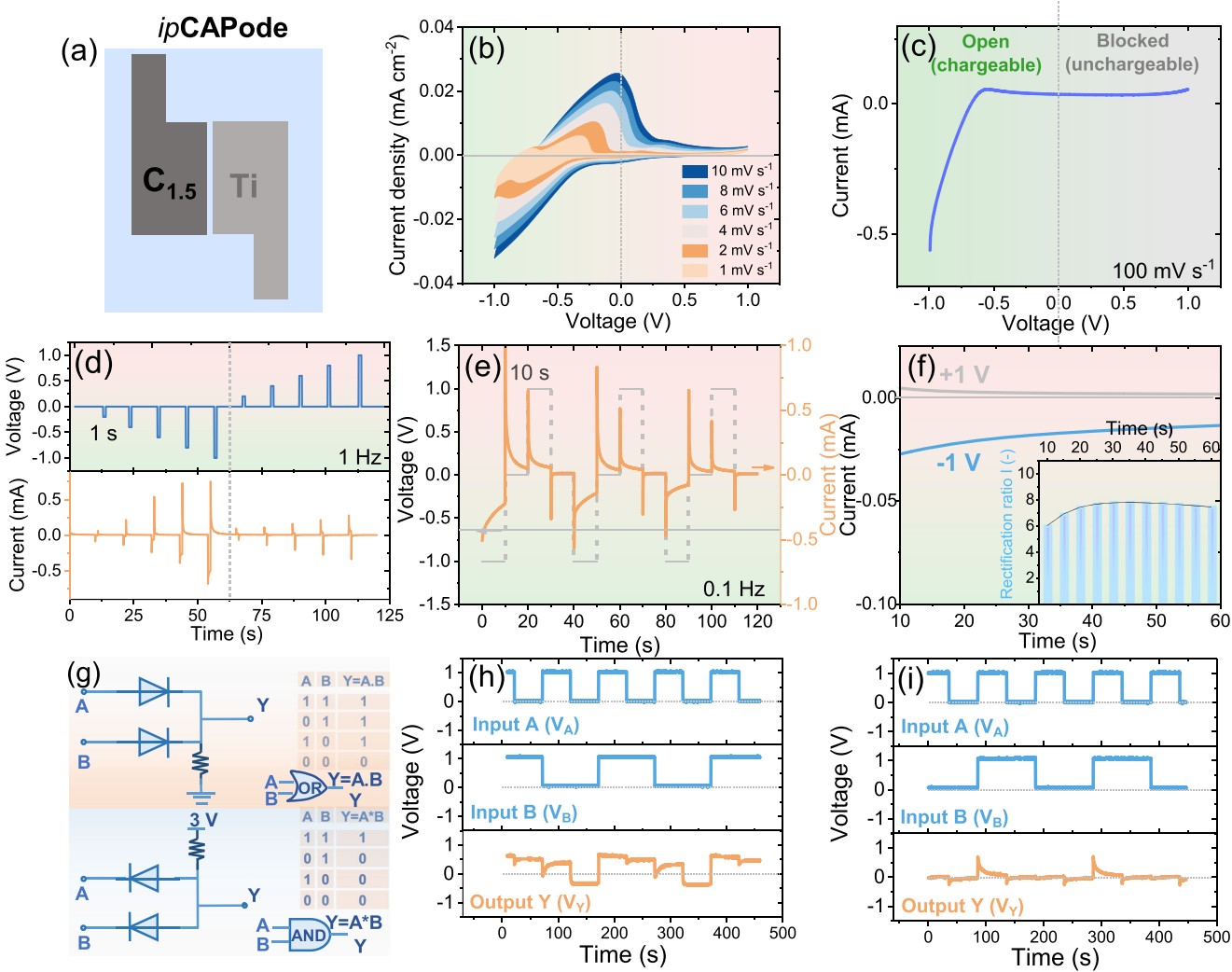

**Fig. 3 | Application of printed CAPode in logic gate circuits. a** The $ip$CAPode architecture. **b** CV and **c** LSV curve of $ip$CAPode. Current response of the $ip$CAPode under **d** 1 s pulse (±0.2 to ±1 V, 1 Hz), **e** 10 s pulse (±1 V 0.1 Hz), and **f** 60 s pulse (±1 V, inset: $RR_I$). **g** Logic circuit scheme for OR (orange) and AND (blue). Input signals and corresponding outputs for **h** OR, and **i** AND circuits.

(10201.30 eV) (Figs. S16f and S17), which confirm the valence change of W species in electrochemical process (details see Section 14, SI)

## Application of CAPode & G-CAPode in logic gate circuits

Screen printing was used to produce micro-devices. $C_{1.5}$ and Ti powders (particle size <100 nm) are used for printing a monolithic $ip$CAPode with planar electrodes and electrolyte configuration (Figs. 3a and S21). The in-plane CAPode ($ip$CAPode) shows a different current response compared with conventional semiconductor diodes[32].

To evaluate the characteristic response of the (Ti | 1 M PWA | $C_{1.5}$) CAPode, a coin cell was assembled and evaluated to comparing and confirming the same logic responses with a printed cell. In Fig. S22, the current of CAPode increases sharply as expected from simple current-voltage ($I–U$) measurements when a "blocked" polarization is applied, whereas it then decreases until it saturates. The current on/off response to a single slow voltage pulse (1 Hz; DC bias from ±0.2 to ±1 V) is presented in Fig. S23. As the voltage increases, the distinct on/off response is visible, which is even more pronounced when the absolute value of the voltage pulse grows. The current-time ($I–t$) curves of the hybrid capacitor diode device at voltages of 1 and −1 V applied alternately (to return their initial states, a 0 V interval was set

before each external bias) are depicted in Fig. S24. The powders of $C_{1.5}$ and Ti (particle size lower than 100 nm) are used for printing a monolithic $ip$CAPode architecture (Figs. 3a and S24), where electrodes and electrolyte are printed on a flat substrate, arranged in a planar configuration. It can be seen that the $ip$CAPode shows different current response characteristics compared to conventional semiconductor diodes[33,34]. We measured key performance metrics such as the on/off current ratio at the voltage of ±1 V (transconductance 1000 cycles) for CAPode (Fig. S25) in the G-CAPode, which keeps more than 84% of $RR_I$ after 1000 cycles.

Compared with a coin cell, the $ip$CAPode presents a much more remarkable current response under "open" polarization (Fig. 3b), which is vital to obtain a high $RR$. The pronounced current response observed in the $ip$CAPode compared to the coin cell device is largely a result of the distinct structural and transport characteristics of the printed design. The $ip$CAPode leverages a porous electrode structure with a higher SSA, which provides more accessible sites for ion adsorption and desorption during operation. Furthermore, the printed architecture of the $ip$CAPode incorporates a shorter ion diffusion path compared to traditional bulk designs in coin cells. This reduced path length minimizes ion transport resistance, allowing ions to move shorter distances and reach the active electrode sites faster.

Specifically, after an "open" polarization is applied, the current increases sharply as expected from simple $I$-$U$ measurements (Fig. 3c), but then decreases until it saturates. Figure 3d shows the current on/off response to a single slow voltage pulse (1 Hz; DC bias from ±0.2 to ±1 V). As the voltage increases, the distinct on/off response is observed, which rises when the absolute value of the voltage pulse increases. Figure 3e shows the current-time ($I$-$t$) curves of the pseudocapacitor diode device at voltages of 1 and −1 V that were applied alternately (to return their initial states, a 0 V interval was set before each external bias). Specifically, the $ip$CAPode exhibits a more rapid decline in current response at the "open" polarization (−1 V) compared to the "blocked" polarization (1 V), when opposite polarizations are applied. This distinct electrochemical behavior is different from the typical current response characteristics observed in conventional semiconductor diodes. The behavior remains similar, but the $RR_I$ is slightly higher than the initial state and becomes steadier at the end (the inset plot in Fig. 3f), consistent with previous observations in the bulk cell (Fig. S25; ~84% retention after 1000 cycles). Faster processes like electron transfer or interfacial reactions dominate the system's dynamics due to $\tau_{device} \ll \tau_{diffusion}$ (Fig. S26). Hence, as previously observed, the $RR$ changes as the duration of the voltage pulse increases, and finally saturates. This deviation in transient current response from conventional diodes is expected and has been discussed in our previous study[18], as well as for comparable reported electrochemical diodes[34]. We note that such a bimodal current response is inherent to an electrolytic cell and will inevitably cause a characteristic behavior of CAPodes differing from conventional diodes.

The logic gate tests effectively demonstrate the feasibility of ion-based computing using the CAPode/G-CAPode system as a key component. Furthermore, achieving miniaturization and integration is essential for advancing scalable ionologic logic gates, enabling new computational paradigms. To demonstrate the potential of capacitive computing with ion-selective redox CAPodes, $ip$CAPodes were assembled to construct OR (the orange areas of Fig. 3g) and AND (the blue areas in Fig. 3g) gate circuits. As illustrated in Fig. 3h, i the input and output signals reflect logic processes that are relatively stable and real-time. Figure 3h demonstrates that the output is low ($V_Y \approx −0.2$ V) only when $V_A$ and $V_B$ are both low ("0" = 0 V), and the output is high ("1" ≈ 0.5 V) as long as one of the inputs is high. Consequently, a simple OR logic has been realized with the ionologic CAPode devices. This output voltage is not constant because in CAPodes, as devices based on hybrid ECs, a slightly different mechanism describes the charge and discharge process. The remarkable negative output voltage, as $V_A$ and $V_B$ are both low, results from the inverse charging of the CAPode connecting $V_B$ due to the generated voltage difference between two CAPodes. When only one of $V_A$ and $V_B$ is high, it is apparent that the output is a bit lower than when $V_A$ and $V_B$ are high both. This is because the corresponding branch (Fig. 3g) is not in a state of open-circuit when $V_A$ or $V_B$ is equal to "0", which results from the discharging of the CAPode as in ECs. Compared to the coin cells, a resistance of 1 kΩ was chosen in $ip$CAPode, while the output voltage of "1" is much lower than 1 V because the resistance and current generated by this circuit are imperfect, inducing a significant IR drop. For AND logic gate based on two $ip$CAPodes, the output is high (≈0.12 V) only when $V_A$ and $V_B$ are both high ("1" = 1 V), and the output is low as long as one of the inputs is low ("0" ≈ 0 V) (Fig. 3i). These results correspond well to the logic function scheme of input signals of (0,0), (1,0), (0,1) and (1,1) in coin cells in Fig. S27. Therefore, all results above validated the value of the $ip$CAPodes system for ion-based computing.

As an additional iontronic element, an $ip$CAPode with gating characteristics ($ip$G-CAPode) was built resembling transistor characteristics (Fig. S28). This 3-electrode architecture integrates a symmetric working CAPode and a gate electrode (GE) that reversibly depletes/injects electrolyte ions into the CAPode channel, enabling precise control over the charge storage capacity and potential of

CAPode electrodes (Figs. 4a and S29)[23]. Under OCV conditions, ions are located at the inter-electrode space of the CAPode (illustrated with a blue background). In the "on" state, the device is effectively charged when the CAPode electrodes are properly polarized, and no bias potential is applied to GE (illustrated with a red background). In the "off" state, a bias potential is applied to GE, resulting in the capture of ions from the inter-electrode space of the CAPode (illustrated with a yellow background). This process leads to capacitance retention in the CAPode. Ti and $C_{1.5}$ powder were used to print an optimized $ip$CAPode as the CAPode and $C_{1.5}$ as the GE in the $ip$G-CAPode structure (Fig. 4b).

The circuit diagram (Fig. 4c) shows applied voltages between GE and two CAPode electrodes. The corresponding CV curves of the different states are shown in Fig. 4d. During cycling of the CAPode (on-stage without applied bias to GE), the CV curve (green solid line) shows expected (Ti | 1 M PWA | $C_{1.5}$) CAPode behavior. The CV curve (solid red line) in the off-state (−1.2 V $vs.$ CE bias applied to GE) is flat, and the capacitive characteristics are effectively suppressed. The cations are immediately depleted in the electrolyte, and the CAPode current decreases due to a deficit of charge carriers (inset in Fig. 4d). The gating characteristics are comparable to those of a field effect transistor or electrochemical transistor when plotting the drain current versus the gate-source voltage[32]. The $I$-$U$ curve (on-stage; dashed green line) perfectly superimposes the initial cycle, demonstrating reversible on/off switchability. The CAPode capacity should recover to nearly 100%. For an aqueous system, if the electrochemical window in Ti | 1 M PWA | $C_{1.5}$ could be widened, the minimum voltage of GE $vs.$ CE (threshold voltage) would be shifted[35,36]. Remarkably, the $RR$ of $ip$G-CAPode is lower than the G-CAPode-based bulky cell (Fig. S30), which may be due to the higher inner resistance of the printed cell.

The $ip$G-CAPode was assembled to construct a logic NOT circuit (inverter), which consists of one $ip$G-CAPode and two 1 kΩ resistors (Fig. 4e). Based on the connection of NOT gate by simulation (Fig. S31), $V_A$ serves as the input signal to the $ip$G-CAPode, and $V_Y$ represents the output signals. As illustrated in Fig. 4f, g, the input and output signals reflect logic processes that are relatively stable and real-time. Figure 4g demonstrates that the output is low ($V_Y \approx 0$ V) when $V_A$ is high ("1" = −1.2 V), and the output is high ("1" = 0.5 V) as long as the input is low. These results correspond well to the logic function scheme of input signals of (0,1), (1,0), for a commercial transistor in Fig. 4f. Consequently, a simple NOT logic gate has been realized with the ionologic $ip$G-CAPode device. It has to be pointed out, that the high output ("1" = 0.5 V) is much lower than 1 V, as the outer resistance, and the high polarization inner resistance of printed structures and current-generated by this circuit are not optimized, resulting in a significant IR drop. Therefore, to obtain distinct output signals ("1" ≈ 1 V and "0" ≈ 0 V), several parameters can be adjusted, including outer resistance, inner resistance, and current generated by this circuit.

Furthermore, the value of capacitive computing with ion-selective redox ionologic devices was demonstrated as CAPode and G-CAPode were successfully integrated to construct a logic NAND gate (Fig. S32; details see in Section 15 is SI).

## Discussion

The proposed asymmetric charge storage system selectively operates with redox-active ions reduced on the Ti electrode, enabling unidirectional current flow. The PWA with Keggin structure was optimized as a low-cost electrolyte, offering high $RR$. Furthermore, the output voltage of transistor-like G-CAPode was demonstrated. Screen-printed CAPodes and G-CAPode were applied to AND, OR, and NOT gates. The resulting ionologic devices can be applied to proof-of-concept integrated circuits (NAND), indicating the potential of capacitive computing with ion-selective redox ionologic devices. In our proof-of-concept devices, the rate performance poses limitations, but downscaling may improve performance indicators in this

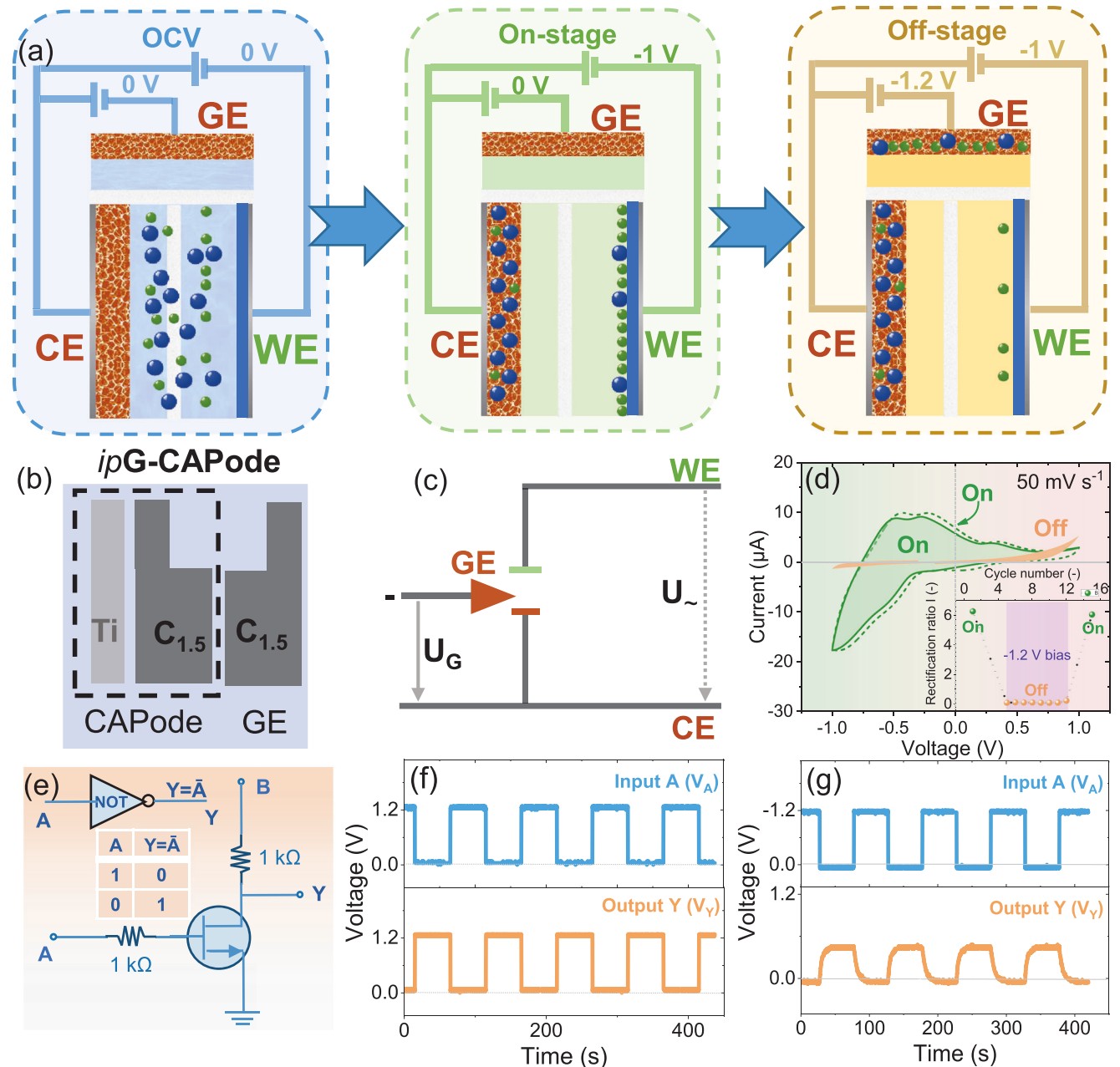

**Fig. 4 | Application of printed G-CAPode in logic gate circuits. a** Schematic of the electroadsorption-based switching mechanism (blue ball: $H_3O^+$ ions; green ball: $PW_{12}O_{40}^{3-}$ ions). **b** Monolithic *ip*G-CAPode architecture. **c** Electrical circuit diagram. **d** *I–U* curves in on/off-state with RR sequence (inset plot). **e** NOT logic circuit scheme. Input and output signals for **f** commercial transistor and **g** *ip*G-CAPode.

respect. Ion-based signal transport rates in axons achieve up to 100 m s⁻¹. Our bioinspired approach parallels the effective operation of nerve systems based on multifarious ions and neurotransmitters, while electronics only operate with electrons and holes. Our concept introducing the general applicability of redox-active electrolytes opens an avenue for the emerging field of ion-based computing.

## Methods

### Reagents and materials
Kynol Europe GmbH® (Germany) supplied the active carbon fibers ACC-5092-10 ($C_{0.7}$), which are based on a material called Novoloid. Pitch-based active carbon fiber A20 ($C_{1.5}$) was supplied by AD'ALL Co® (Japan). Carbon Molecular Sieve (CMS; $C_{0.55}$) was purchased from Guangdong New Energy Technology Co., Ltd.® (China). Carbon black AB208928 (50% compressed; 99,9%) was purchased from ABCR GmbH

& Co® (Germany). Polytetrafluoroethylene (PTFE) was purchased from abcr GmbH - Karlsruhe, Germany. Polyvinylidene Fluoride (PVDF; ≥99.5) was purchased from MTI Corporation. N-methyl-2-pyrrolidone (NMP; ACS reagent; ≥99.0%), Titanium powder (Ti powder; 99.98%), and phosphotungstic acid hydrate ($H_3[PW_{12}O_{40}]$·$xH_2O$; product number: P4006; ≤100%) were purchased from Sigma Aldrich® (USA). Glass fiber separators GF/D were purchased from Whatman® (UK). Carbon black Super p MA-EN-CO-01 and titanium mesh (>99.5% of Ti; as a rigid solid electrode) were purchased from Canrd® (China). All chemicals were utilized exactly as they came into the chemical shipment without further purification.

### Screen-printing design
Two screens were used, equipped with either a standard PET mesh (150/31; 22.5°) and coated by hand with an all-round photoemulsion

(FLX Screen; Siebdruckversand; China). Printed structures are simple lines with line lengths of 12 mm and widths 3.25 mm, and the weight of each electrode is around 3 mg. Furthermore, interdigitated structures with line width and gaps of 100 μm were designed to print micro-ionologic devices. Interdigitated structures with 50 μm gap, which is close to the resolution of the screen-printing method (25–30 μm), can be printed, but the number of overprint passes can hardly exceed 2 (since by three spreading of the ink, fingers will merge). Hence, larger gaps (e.g., 100–200 μm) are preferred for overlayer printing. The existence of large particles in the sediment ink is another limiting factor for increasing the print resolution.

## Characterization of carbon materials

Nitrogen physisorption was measured volumetrically at 77 K on a BELSORP apparatus (Microtrac MRB®). The AUTOSORB-iQ-C-XR from Quantachrome® was used to perform a physisorption study with argon at 87 K. The specific surface area (SSA) was determined using the Brunauer-Emmett-Teller (BET) method, and the total pore volume was calculated using the cumulative results according to the density functional theory (DFT) for the partial pressure range of $0.05 < P/P_0 < 0.20$. The total pore volume up to 2 nm was used in DFT calculations to derive micropore volumes. The samples were degassed at 150 °C for at least 12 h prior to measurement.

$^{31}$P magic angle spinning nuclear magnetic resonance (MAS NMR) spectra were obtained using a commercially available solid-state 2.5 mm double resonance ($^1$H, X) MAS NMR probe in a Bruker Avance Neo spectrometer (Bruker Biospin; Karlsruhe; Germany) tuned to an operating frequency of 121.49 MHz. The sample rotation was set to 15 kHz, and chemical shifts were referenced to TMS using phosphorylated serine as a secondary reference (0.3 ppm relative to TMS). Sample loading was performed by the incipient wetness method in an Ar-filled environment ($x < 0.1$ ppm $H_2O$ and $O_2$). Carbon was loaded with a solution of 1 M PWA in deuterated water (d-$H_2O$) corresponding to two times the pore volume, as determined from the $N_2$-adsorption isotherms. The resulting mixture was mortared for 120 s. For the carbon material, three signals occur, which are assigned to ions in the free bulk (1), in contact with the outer particle service (2), and adsorbed inside the pores (3), see also the inserted signal decomposition and the sketch (the inset plot in Fig. 2e).

## Electrode slurry preparation

The carbon slurry was prepared by mixing 80 wt% $C_{1.5}$ as the active material, 10 wt% Super P (MA-EN-CO-01) in NMP as the solvent to as a conductive agent, and 10 wt% PVDF as a binder to form a homogeneous ink. The mixture was processed using ball milling for 60 min to ensure uniform dispersion, break down agglomerates, and improve particle contact for better electrode performance. This slurry was coated onto Ti mesh as a CE for parafilm-fixed cells to assess long-term cycling performance and used in a UV tube volume cell for in situ Raman and UV measurements. Additionally, the slurry was employed for screen-printed electrodes, while titanium powder-based electrodes were prepared using the same process as the carbon slurry.

## Free-standing carbon film electrode preparation

The carbon electrodes were prepared by mixing 85 wt% $C_{1.5}$ as the active material, 10 wt% carbon black (AB208928) as a conductive agent, and 5 wt% PTFE as a binder, specifically using a 10 wt% PTFE aqueous dispersion, to ensure uniform distribution and mechanical integrity. The components were thoroughly mixed to form a homogeneous paste, which was then processed by rolling to achieve a free-standing film electrode with a controlled thickness of $100 \pm 20$ μm. To create electrodes suitable for electrochemical testing, some of the films were punched into discs with a diameter of 10 mm. For activation

and removal of residual solvents or moisture, the electrodes were stored under a vacuum at 333 K for 24 hours, which enhances their structural stability and electrochemical performance.

## Two-electrode cell setups

A homemade cell (Fig. S3a), constructed from polyether ether ketone with a two-electrode arrangement, was used for 2-electrode electrochemical testing. The electrode discs, which had the same mass for both the WE and CE (approximately $5 \pm 1$ mg), were pressed onto titanium current collectors that were coated with carbon ink (DAG EB-012). Glass microfiber filters (Whatman; $\varnothing = 12$ mm) served as the separator between the electrodes. After the cell was assembled, it was filled with 100 μL of the respective electrolyte, ensuring a uniform environment for electrochemical measurements.

A parafilm-sealed cell (active area: working electrode (WE; coated with the carbon slurry (mass loading: $6 \pm 1$ mg) 0.2 cm × 2 cm, CE (pure Ti mesh) 1 cm × 2 cm) was employed for long cycling electrochemical measurements (Fig. S3b). Glass microfiber filters (Whatman, 1 cm × 2 cm) served as the separator between the electrodes, ensuring effective ionic conductivity while minimizing side reactions. After the assembly, the cell was filled with 100 μL of the respective electrolyte. This configuration allows for the investigation of long-term cycling stability and the evaluation of charge-discharge behaviors under controlled conditions.

For in situ Raman and UV measurements, a tube volume cell (active area: WE 0.1 cm × 2 cm; CE 0.5 cm × 2 cm) was employed (Fig. S3c). The mass loading of carbon was also $6 \pm 1$ mg for the electrodes in this setup. The electrolyte used was 2 mL of 1 M $H_3PW_{12}O_{40}$. The rubber stopper is used to fix the electrodes and seal the system in the setup.

For operando X-ray synchrotron diffraction measurements, in situ CR2025 coin cells (Fig. S3d) with Kapton windows were utilized, enabling real-time diffraction studies during electrochemical reactions. Glass microfiber filters (Whatman; $\varnothing = 12$ mm) were used as the separator to ensure effective ion transport between the working and CEs while minimizing diffraction interference. After the coin cells were assembled, they were filled with 100 μL of the respective electrolyte.

## 3-electrode cells for monitoring potential of WE and CE

Based on the above-mentioned homemade 2-electrode cell, a Hg/HgSO$_4$/saturated K$_2$SO$_4$ ($E = 0.642$ V vs. SHE) reference electrode was inserted in the electrolytes through the hole of the cell (Fig. S3a). The Hg/HgSO$_4$ electrode was calibrated against a standard Ag/AgCl/3 M NaCl reference electrode ($E = 0.197$ V vs. SHE), with a measured potential of 0.443 V vs. Ag/AgCl by OCV measurement. The calibration was performed in a 3 M NaCl aqueous solution, with a total volume of 20 mL in the electrochemical cell. No correction for solution resistance was applied, as the cell design and operating conditions were optimized to minimize resistance-related effects. This calibration confirms that the potential of the Hg/HgSO$_4$ electrode is accurate and consistent with standard reference values. The potentials of working and CE (vs. Hg/HgSO$_4$ reference) were monitored and recorded at the same time using a Biologic VMP-3 potentiostat.

## Electrochemical measurements

We used a VMP300 from BioLogic with CV, galvanostatic cycling with potential limitation, and electrochemical impedance spectroscopy to investigate the electrochemical performance. Prior to data analysis, the electrode was cycled four times to stabilize its electrochemical performance. Each measurement was repeated 2 times, and the final data presented in the figures related to quantifying the performance of CAPode and G-CAPode are based on the mean value of these three repetitions. The electrochemical measurements were conducted in a controlled laboratory environment. The temperature was maintained at an average of $293 \pm 5$ K throughout the experiments.

### Characterization of mechanism

In situ measurement: DXR SmartRaman Spectrometer (532 nm laser) was applied to collect Raman spectra. In situ cylindrical quartz glass tube cell was prepared by inserting one WE (Ti mesh with 0.5 cm × 2 cm) and one CE (10 ± 2 mg carbon coated on Ti mesh with 0.5 cm × 2 cm, both separated by a rubber stopper). After assembly, the cell was filled with ≈300 μL of the respective electrolyte (1.5 cm). IVIUM pocketSTAT 2 potentiostat was connected to the in situ tube cell for electrochemical testing. The different potentials (0, −0.4, −0.5, −0.6, −0.7, −0.8, −0.9, and −1 V) were applied to the cell, and each potential was held for 600 s. Raman spectra were collected in real-time during potential cycling, with each spectrum registered for 120 s.

Operando X-ray synchrotron diffraction measurements were performed at beamline BL04 at the ALBA cells synchrotron (Spain), while ex-situ XAS were performed at the P64 beamline at PETRA III, German Electron Synchrotron (Hamburg). For measurements, dedicated in situ CR2025 coin cells with Kapton windows were used. The prepared free-standing $C_{1.5}$ carbon film is used as the working electrode, with a Ti mesh of the same size serving as the CE. A 1 M PWA aqueous solution (100 μL) is used as the electrolyte.

### Data availability

The data that support the findings of this study are openly available in a public repository that issues datasets with DOIs. The DOI to access the Raw data are: https://doi.org/10.5281/zenodo.15296075.

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

## Acknowledgements

The authors acknowledge the funding received from the European Research Council (ERC) under the European Union's Horizon 2020 Research and Innovation Program (grant agreement no. 101054940; title: Ultracapacitor Logic Gates). The authors thank Kynol® GmbH for providing active carbon fiber. Hanfeng Zhou thanks China Scholarship Council (CSC) for financial support. We appreciate the HighPerformance Computing Center of Shanghai University, and Shanghai Engineering Research Center of Intelligent Computing System (no. 19DZ2252600) for providing the computing resources and technical support.

## Author contributions

H.Z.: conceptualization; methodology, investigation; writing–original draft; visualization. P.G.: writing–review & editing; supervision; visualization. T.Z.: DFT calculation; revision. P.L.: guidance for in situ Raman and UV; revision. X.Z.: simulation of logic gates; revision. C.L.: XAS measurement and data analysis; revision. J.K.: NMR measurement and data analysis; revision. K.H.: guidance of the tests for logic gates; revision. Y.L.: XRD measurement and structure modification of Keggin; revision J.Q.: ex situ Raman measurement and data analysis; revision. A.B.: guidance of the connection of G-CAPode; revision P.X.: XRD measurement and data analysis; revision. J.G.: writing–review & editing; supervision. D.M.: XAS measurement and data analysis; revision S.C.B.M.: guidance of the tests for logic gates; revision. E.B.: NMR measurement and data analysis; S.K.: conceptualization; funding acquisition; writing–review & editing; supervision.

## Funding

## Competing interests

The authors declare no competing interests.
