## [Transparent Peer Review file · Nature Communications]

Redox-active Keggin-type electrolyte based printed ionologic devices

Corresponding Author: Professor Stefan Kaskel

Version 0:

Reviewer comments:

Reviewer #1

(Remarks to the Author)

Review of "Redox-active Keggin-type electrolyte-based printed ionologic devices":

This paper explores the application of a redox-active Keggin-type electrolyte in printed ionologic devices, focusing on its potential for electrochemical capacitor diode (CAPode) applications. Unlike prior studies that have reported symmetric electrochemical capacitor diodes, this work introduces an asymmetric electrode structure that combines redox-active electrolytes. By exploiting the differing current and capacitance responses under positive and negative potentials in the asymmetric device, the authors achieve functional capacitor diode behavior. This approach broadens the structural possibilities for electrochemical capacitor diodes and demonstrates a promising advancement for iontronic applications. I recommend that the paper be accepted only after major revisions to address critical issues.

Major Revisions Needed:

1. The manuscript highlights the use of Keggin-type electrolytes but lacks a clear explanation for this selection. Given that other redox electrolytes discussed in the manuscript display better rectification ratios, the authors are encouraged to provide a detailed rationale for selecting the Keggin-type electrolyte over these alternatives.
2. While the data presented is thorough, the manuscript's overall quality suffers due to formatting and logical flow issues. Several typographical and logical errors, such as duplicated sections (e.g., two "2.2" sections), need correction to ensure coherence.

Minor Revisions Needed:

1. While the importance of ion transistors is emphasized, a clearer distinction between ion transistors and electrochemical capacitor diodes is needed. The manuscript should clarify the advantages and unique aspects of CAPodes in comparison to ion transistors to avoid ambiguity for readers.
2. On Page 4, the authors describe using a plane Ti electrode, yet references to a Ti mesh or $\text{TiO}_2@Ti$ appear in other sections and supporting documents. Consistent terminology and clarification are required throughout the paper. Specifically: in Fig. S7(e) and elsewhere in the manuscript, both Ti and $\text{TiO}_2@Ti$ are used interchangeably. This discrepancy should be rectified.
3. Fig. 1a: This figure appears to draw directly from "Advanced Materials, 35(25), 2301218" without additional discussion regarding neurotransmission or ion computing applications. It is suggested that this figure be removed unless relevant context or application in this manuscript is provided.
4. Fig. 1b and 1c: The central image in Fig. 1b lacks an adequate description, and Fig. 1c remains not explained within the text.
5. On Page 6, line 130, the notation "Ti | redox electrolyte | carbon@metal" unnecessarily highlights "metal" as a current collector. Simplifying this notation would reduce redundancy.
6. In Line 138, emphasizes the presence of TiO_2 on the Ti mesh, raising questions regarding its specific role. The authors should clarify if the redox reaction occurs primarily on the TiO_2 surface and whether the electrode should be labeled $\text{TiO}_2@Ti$ consistently.
7. Line 215: The manuscript refers to "open circuit voltage (OCV)" in Fig. 1a, which is labeled as "OVC" and should be corrected.
8. Fig. 2a and associated text: The color coding lacks consistency; for example, red and blue are used for C and Ti electrodes in Fig. 2a 2b, respectively, but the roles are reversed in Fig. 2c.
9. In Fig. 2a, Ti's GCD curve is truncated. The rationale for recalculating the scan rates for Ti and C in Fig. 2b, along with the methodology behind this calculation, requires clarification.
10. Line 249: The statement seems to refer to "the asymmetric cell as in Fig. 2a."

11. In Lines 276-284, the authors discuss the influence of electrode pore size on rectification ratios. This discussion should be expanded to include the effects of the pore size of the carbon electrode on redox reactions occurring at the Ti electrode in CAPodes.
12. As previously commented in the major revision, additional interpretation of Figs. 2g-2i is necessary, including any observations related to different electrolytes' performance.
13. Additional experimental details are needed for in-situ Raman tests presented in Figs. 3a-3c.
14. The discussion of rectification mechanisms in Fig. 3 is well-supported by advanced in-situ techniques, yet the study lacks innovative insights into the redox reaction mechanism. The author should focus more on the asymmetric mechanisms in CAPodes, such as further exploring the role of the C electrode in triggering Ti-side redox reactions.
15. In line 402, the authors mention using Ti powder for printed electrodes. Clarification is needed on how this material differs from the other Ti materials discussed.
16. Including an actual image of the ipCAPode or NAND logic circuit setup would enhance the paper's impact.
17. What is the impact of the electrochemical window on NAND logic circuits or ipG-Cap architecture?
18. Could you provide the cycle stability of CAPodes in this structure as NAND logic circuits or in an ipG-Cap architecture?
19. It is recommended that the authors discuss limitations on practical applications of the electrochemical capacitor diode and explore its potential future developments.

Reviewer #2

(Remarks to the Author)

This paper introduces an innovative asymmetric electrochemical charge storage system, termed CAPode, which selectively interacts with redox-active ions to achieve unidirectional charge storage and exhibits rectifying characteristics. By controlling different redox couples, the operating voltage range can be adjusted. The selection of 1M phosphotungstic acid as the optimized electrolyte demonstrates high rectification ratios and low costs. Experimental validation shows the potential application of CAPode in the integration of logic gates, highlighting the promise of ion-logic devices in integrated circuits. The paper is clearly written with rigorous logic, making a significant contribution to the fields of electrochemical charge storage and logic operations, and exhibiting a high level of innovation and academic value. Hence, I recommend the manuscript to be published with major revisions addressed. The comments are as follows which would improve the quality of the manuscript:

1. The authors introduced ionic logic devices in the introduction, it is essential for the author to introduce the preliminary concept of ion electronics. Relevant articles to reference include: National Science Review 2024, nwae322; Materials Today 2024, 74, 187-202.
2. An electrolyte 1 M phosphotungstic acid (H₃PW₁₂O₄₀; PWA) with Keggin structure, for this electrolyte, the reason why this electrolyte was chosen and what are the advantages of this electrolyte for constructing supercapacitor diodes. This electrolyte is not widely available, what is its cost and it is necessary for the authors to explain this.
3. As shown in Figure 2i, the rectification ratio of the supercapacitor diode based on Zn(TFSI)₂ is also quite high. The authors could present the performance of this device.
4. As shown in Figure 3b, when the voltage drops to -1V, the Raman peak intensity significantly decreases, and the authors should explain this phenomenon. Additionally, is it necessary to perform normalization? Furthermore, the authors did not present the changes in the Raman peaks under positive voltage conditions.
5. Notice that the authors only show performance up to 100 mV s⁻¹ scan rate. In fact, one of the most critical factors for ion devices is the rate of ion motion, and it is necessary for the authors to show CV curves for higher scan rates.
6. The series of logic gates developed by the authors represents a significant breakthrough in the application of supercapacitor diodes. The authors could explore the advantages of ion-integrated circuits based on these constructions, as well as potential future application scenarios. For example, energy storage chips. Relevant articles to reference include: Materials Today 2024, 74, 187-202.

Reviewer #3

(Remarks to the Author)

The authors present the CAPode, an asymmetric electrochemical charge storage device that uses selective redox reactions on a Ti electrode for one-way charge storage. This system is adjustable within a specific voltage range and exhibits rectification characteristics, which make it highly promising for integration into logic gates and potential capacitive computing applications with ion-selective redox devices.

Overall, I find it an interesting and creative approach to use redox chemistry and asymmetric electrode geometry to achieve current rectification. The concept is well-presented, though I believe the manuscript could be strengthened if the authors more clearly distinguished the novelty of this work compared to their previous reports and recent literature on redox gating and organic electrochemical transistors. It would be helpful to highlight what aspects are unique in this study that advance the field beyond existing literature. For instance, they might consider clarifying the differences from key prior works such as Lochmann et al., 2020 (Adv. Funct. Mater.), Gellrich et al., 2024 (Adv. Mater.), and others listed.

S. Lochmann, Y. Bräuniger, V. Gottsmann, L. Galle, J. Grothe, S. Kaskel, Switchable Supercapacitors with Transistor-Like Gating Characteristics (G-Cap). Adv. Funct. Mater. 2020, 30, 1910439. <https://doi.org/10.1002/adfm.201910439>

C. Gellrich, L. Shupletsov, P. Galek, A. Bahrawy, J. Grothe, S. Kaskel, A Precursor-Derived Ultramicroporous Carbon for

Printing Iontronic Logic Gates and Super-Varactors. *Adv. Mater.* 2024, 36, 2401336.
<https://doi.org/10.1002/adma.202401336>

A. Bahrawy, P. Galek, C. Gellrich, J. Grothe, S. Kaskel, Advanced Redox Electrochemical Capacitor Diode (CAPode) Based on Parkerite (Ni₃Bi₂S₂) with High Rectification Ratio for Iontronic Applications. *Adv. Funct. Mater.* 2024, 34, 2405640.
<https://doi.org/10.1002/adfm.202405640>

E. Zhang, N. Fulik, G.-P. Hao, H.-Y. Zhang, K. Kaneko, L. Borchardt, E. Brunner, S. Kaskel, *Angew. Chem. Int. Ed.* 2019, 58, 13060.

P. Tang, W. Tan, F. Li, S. Xue, Y. Ma, P. Jing, Y. Liu, J. Zhu, X. Yan, A Pseudocapacitor Diode Based on Ion-Selective Surface Redox Effect. *Adv. Mater.* 2023, 35, 2209186. <https://doi.org/10.1002/adma.202209186>

Ma, H., Liang, J., Qiu, J., Jiang, L., Ma, L., Sheng, H., Shao, M., Wang, Q., Li, F., Fu, Y., Wang, J., Xie, E., Chai, Y. and Lan, W. (2023), A Biocompatible Supercapacitor Diode with Enhanced Rectification Capability toward Ion/Electron-Coupling Logic Operations. *Adv. Mater.*, 35: 2301218. <https://doi.org/10.1002/adma.202301218>

L. Zhang, C. Liu, H. Cao, A. J. Erwin, D. D. Fong, A. Bhattacharya, L. Yu, L. Stan, C. Zou, M. V. Tirrell, H. Zhou, W. Chen, Redox Gating for Colossal Carrier Modulation and Unique Phase Control. *Adv. Mater.* 2024, 36, 2308871.
<https://doi.org/10.1002/adma.202308871>

Hui Cao, Changjiang Liu, Dillon D. Fong, Anand Bhattacharya, Matthew V. Tirrell, Hua Zhou, Wei Chen; Redox gating-induced modulation of charge carrier density and lattice expansion in LaNiO₃ thin films. *Appl. Phys. Lett.* 12 August 2024; 125 (7): 074101. <https://doi.org/10.1063/5.0217899>

Ahsan Raza, Umar Farooq, Khalida Naseem, Sarfaraz Alam, Mohammad Ehtisham Khan, Akbar Mohammad, Waleed Zakri, Muhammad Yasir Khan, (2024), A focused review on organic electrochemical transistors: A potential futuristic technological application in microelectronics, *Microchemical Journal*, 207, 111737.
<https://doi.org/10.1016/j.microc.2024.111737>.

Rivnay, J., Inal, S., Salleo, A. et al. Organic electrochemical transistors. *Nat Rev Mater* 3, 17086 (2018).
<https://doi.org/10.1038/natrevmats.2017.86>

Guo, J., Chen, S.E., Giridharagopal, R. et al. Understanding asymmetric switching times in accumulation mode organic electrochemical transistors. *Nat. Mater.* 23, 656–663 (2024). <https://doi.org/10.1038/s41563-024-01875-3>

Huang, W., Chen, J., Yao, Y. et al. Vertical organic electrochemical transistors for complementary circuits. *Nature* 613, 496–502 (2023). <https://doi.org/10.1038/s41586-022-05592-2>

The technical characterizations presented, such as electrical measurements and X-ray/spectroscopy studies, are well done. However, it is surprising that the manuscript doesn't include a single real photograph of the device, nor any SEM or morphological images. For instance, Figures 1, 4a, and S24 are helpful schematics, but readers would benefit from seeing actual images of the device, as this might also help explain aspects like the "high inner resistance of printed cell compared to coin cell" mentioned on Line 512. Including images of printed patterns could add clarity.

There are a few areas where the figure quality could be improved. For example, Figure S1 is challenging to interpret due to low inset figure readability. Improving resolution or font size in these areas would enhance readability. Additionally, a few minor textual improvements are needed, such as moving the abbreviation "SSA" in Line 117 to the correct location.

It might also be helpful to revise the structure of some sections for better clarity. For example, on Line 187, discussions around SI content could be streamlined by either moving the equations and parameters to the SI entirely or bringing the related discussion into the main text, as appropriate. "Rectification ratio" is referenced frequently throughout but lacks a clear definition in the main text.

In Figure 3f, the XANES data for tungsten could benefit from additional data points below the L3 absorption band. Recording only ~50 eV below the band may limit the modeling accuracy and affect the baseline adjustment. Also, in the SI, there's a lack of detail on data processing for this XANES data—providing specifics would add transparency.

Additionally, the manuscript could improve by explaining certain conclusions more clearly. For example:

Line 359: The authors refer to the W K3 edge, which I believe should be the L3 edge, as the W K3 edge is around 69.5 keV.

Line 383: This section could benefit from more interpretation. What is the main conclusion, or purpose, of this experiment?

Line 405: When discussing the device's time response, quantifying or fitting a time constant and comparing it to ion diffusion values could clarify how ion dynamics impact performance.

Line 423: Why does the ipCAPode show a more pronounced current response than the coin cell?

In Figure 5, it is a bit unclear if the gate electrode (GE) is modulating the CE-WE current by changing the ion density or if another effect might be responsible, such as a transverse electric field or larger surface area effects. Additionally, the schematic in Fig. 5a does not match Fig. 5d, unless the GE is indeed placed on top of the channel. More clarity here would

improve reader comprehension.

In general, I recommend reconsideration after a major revision to address these points. Thank you for the opportunity to review this work, and I look forward to seeing how it develops with these refinements.

Version 1:

Reviewer comments:

Reviewer #1

(Remarks to the Author)

Thank you very much for your response to my previous comments. The manuscript has improved significantly. The authors have addressed my comments point by point, and the additional data and further analysis provide strong support for the experimental results. I only have 2 minor points as follows:

1.As mentioned in the abstract, the most innovative aspect of this study lies in the fact that "this unidirectional capacity is achieved through asymmetric polarization between a plane metal and a porous carbon electrode, enabling selective redox reactions on the metal surface." However, considering the current title of the manuscript, should it be more specific, such as "Redox-active Keggin-type electrolyte" be noted in abstract, or would "Redox-active electrolyte based printed ionologic devices" be a more suitable option as title?

2.Regarding Comment 20, the device presented in this manuscript does not appear to exhibit good cycling stability, especially when compared to the similar work previously published by the authors' research group (*Advanced Functional Materials* 34.45 (2024): 2405640). For the device in this study, what are the primary factors limiting its cycling stability? Could the authors provide further discussion on this aspect?

Reviewer #2

(Remarks to the Author)

I have read all the replies the author answered in the review report. The author addressed all my concerns in great detail, and the revised manuscript is suitable for the publication. I suggest the publication of this manuscript at the current version.

Reviewer #3

(Remarks to the Author)

Thank the authors for carefully considering my comments and addressing them thoughtfully. I appreciate the effort you have put into refining the manuscript. The revisions have significantly strengthened the clarity and impact of your work. I fully support the publication of this manuscript in *Nature Communications*.

Open Access This Peer Review File is licensed under a Creative Commons Attribution 4.0 International License, which permits use, sharing, adaptation, distribution and reproduction in any medium or format, as long as you give appropriate credit to the original author(s) and the source, provide a link to the Creative Commons license, and indicate if changes were

made.

Reviewer #1:

“This paper explores the application of a redox-active Keggin-type electrolyte in printed ionologic devices, focusing on its potential for electrochemical capacitor diode (CAPode) applications. Unlike prior studies that have reported symmetric electrochemical capacitor diodes, this work introduces an asymmetric electrode structure that combines redox-active electrolytes. By exploiting the differing current and capacitance responses under positive and negative potentials in the asymmetric device, the authors achieve functional capacitor diode behavior. This approach broadens the structural possibilities for electrochemical capacitor diodes and demonstrates a promising advancement for iontronic applications. I recommend that the paper be accepted only after major revisions to address critical issues.”

Dear Reviewer #1,

We sincerely thank you for your positive review and thoughtful comments on our manuscript. In response to your recommendation for major revisions, we carefully addressed the critical issues you raised to ensure our findings were presented as clearly and comprehensively as possible. We also expanded upon the functional advantages of our asymmetric electrode structure in enabling capacitor diode behavior, as well as refined the discussion to further highlight its potential in iontronic applications. We believe that with these revisions, the paper will provide a valuable contribution to the field.

Comment 1: *“The manuscript highlights the use of Keggin-type electrolytes but lacks a clear explanation for this selection. Given that other redox electrolytes discussed in the manuscript display better rectification ratios, the authors are encouraged to provide a detailed rationale for selecting the Keggin-type electrolyte over these alternatives.”*

Thank you for your comment, which is essential for the overall narrative of this paper. Now we concluded the advantages of using a Keggin-type electrolyte, which mainly lies in the high reversibility and defined redox window, to accurately reflect this distinction with other redox electrolytes and added reference related. We have added a short description in the

manuscript (due to word constraints required by Nature Communications) highlighting the advantages of using PWA in our research, but a more detailed discussion was included in the SI.

Please see the highlighted changes in lines 282-288 of the Manuscript, and lines 400-455 of SI.

Comment 2: *“While the data presented is thorough, the manuscript’s overall quality suffers due to formatting and logical flow issues. Several typographical and logical errors, such as duplicated sections (e.g., two “2.2” sections), need correction to ensure coherence.”*

We understand the importance of formatting and logical coherence in presenting our data effectively. We have reviewed and corrected formatting inconsistencies, including the duplicated section numbering (notably the two "2.2" sections). These changes have been applied to ensure a more logical flow and clarity.

Please see the highlighted changes in line 167 and 297 of the Manuscript.

Comment 3: *“While the importance of ion transistors is emphasized, a clearer distinction between ion transistors and electrochemical capacitor diodes is needed. The manuscript should clarify the advantages and unique aspects of CAPodes in comparison to ion transistors to avoid ambiguity for readers.”*

We agree that a clearer distinction between ion transistors and electrochemical capacitor diodes (CAPodes) is essential for enhancing the clarity of the manuscript and avoiding ambiguity for readers. To address this, we have revised the introduction section and included **table S3** to explicitly highlight the differences between these two devices.

Please see the highlighted changes in lines 59-63 of the Manuscript and lines 791-795 of the SI.

Comment 4: “On Page 4, the authors describe using a plane Ti electrode, yet references to a Ti mesh or $\text{TiO}_2@\text{Ti}$ appear in other sections and supporting documents. Consistent terminology and clarification are required throughout the paper. Specifically: in Fig. S7(e) and elsewhere in the manuscript, both Ti and $\text{TiO}_2@\text{Ti}$ are used interchangeably. This discrepancy should be rectified.”

Thank you for pointing out this inconsistency. We have revised the text throughout the manuscript and supporting documents to ensure consistent terminology. We now use "Ti" exclusively to refer to the titanium electrode in its metallic form. The reason why this TiO_2 was used in the manuscript somewhere is that we used it for DFT calculation in SI, which required the surface information of the electrode. However, the surface TiO_2 is not important to achieve the rectification effect of CAPode. The redox reaction of $\text{W}^{6+}/\text{W}^{5+}$ happens in the CAPode system of Ti | electrolyte | carbon, which is achieved through asymmetric polarization between a plane metal with low SSA and a porous carbon electrode with high SSA, enabling selective redox reaction on the metal electrode, resulting in the controllable orientation of unidirectional capacity. In **Fig. S7e** and other relevant sections, we have clarified these terms to reflect this distinction accurately. This change should now provide a clearer understanding and avoid any confusion for the reader.

Please see the highlighted changes in line 214, 349-350, 354-356, 618-619, and 741-744 of the SI.

Comment 5: “Fig. 1a: This figure appears to draw directly from "Advanced Materials, 35(25), 2301218" without additional discussion regarding neurotransmission or ion computing applications. It is suggested that this figure be removed unless relevant context or application in this manuscript is provided.”

Fig. 1a was included to provide foundational background on the structural design pertinent to ion transport mechanisms, which is better for the TOC. We removed **Fig. 1a** from the manuscript as suggested, to maintain focus on the new findings presented in this work.

Please see the highlighted changes in lines 105-110 of the Manuscript.

Comment 6: *“Fig. 1b and 1c: The central image in Fig. 1b lacks an adequate description, and Fig. 1c remains not explained within the text.”*

Thank you for pointing out the need for clearer descriptions of **Fig. 1b, c**, which is **Fig. 1a, b** now. We have added a paragraph describing the redox process illustrated in this Figure.

Please see the highlighted changes in lines 135-146 of the Manuscript.

Comment 7: *“On Page 6, line 130, the notation “Ti | redox electrolyte | carbon@metal” unnecessarily highlights “metal” as a current collector. Simplifying this notation would reduce redundancy.”*

We agree that the notation can be simplified to improve clarity. Throughout the manuscript, we have revised it to "Ti | redox electrolyte | carbon material (C_{1.5}/C_{0.70}/C_{0.55})" throughout the manuscript and removed the redundant reference to "metal" as a current collector.

Comment 8: *“In Line 138, emphasizes the presence of TiO₂ on the Ti mesh, raising questions regarding its specific role. The authors should clarify if the redox reaction occurs primarily on the TiO₂ surface and whether the electrode should be labeled TiO₂@Ti consistently.”*

Thank you for raising this important point. The redox reaction of W⁶⁺/W⁵⁺ happens in the CAPode system of Ti | electrolyte | carbon, which is achieved through asymmetric polarization between a plane metal with low SSA and a porous carbon electrode with high SSA, enabling selective redox reaction on the metal electrode, resulting in the controllable orientation of unidirectional capacity. Although different materials can have various overpotentials, the surface structure (thin-layer TiO₂) on the Ti electrode (99.95% of Ti) just can affect slightly its polarization, but it is not decisive for achieving the rectification effect. We used other low surface area materials (i.e., pure glassy carbon (diameter: 2.5

mm) (**Fig. A1**); W wire, thickness: 0.25 mm (**Fig. A2**) as WE and high surface area carbon as CE, and they also function like Ti mesh, which also achieved a good CAPode performance. Therefore, the surface structure (thin-layer TiO₂) on the electrode is not an essential feature and can be omitted. Therefore, we chose to use "Ti" exclusively to refer to the Ti electrode in its metallic form for the whole manuscript and SI.

Fig. A1. CV curves for 6 cycles at the same scan rate (100 mV s^{-1}) of asymmetric cell (glassy carbon | 1 M PMA | C_{1.5}).

Fig. A2. CV curves for 6 cycles at the same scan rate (100 mV s^{-1}) of asymmetric cell (W wire | 1 M PMA | C_{1.5}).

Comment 9: "Line 215: The manuscript refers to "open circuit voltage (OCV)" in Fig. 1a, which is labeled as "OVC" and should be corrected."

Thank you for noticing this labeling error. We apologize for the oversight. We have corrected the label in **Fig. 1a** to "OCV" to match the term "*open circuit voltage*" used in the text.

Please see the highlighted changes in line 189 of the Manuscript.

Comment 10: *"Fig. 2a and associated text: The color coding lacks consistency; for example, red and blue are used for C and Ti electrodes in Fig.2a 2b, respectively, but the roles are reversed in Fig. 2c."*

Thank you for highlighting this inconsistency in the color coding. To maintain clarity and avoid confusion, we have standardized the color scheme across all parts of **Fig. 2**, ensuring that red consistently represents the C (carbon) electrode and blue consistently represents the Ti electrode.

Please see the highlighted changes in line 189 of the Manuscript.

Comment 11: *"In Fig. 2a, Ti's GCD curve is truncated. The rationale for recalculating the scan rates for Ti and C in Fig. 2b, along with the methodology behind this calculation, requires clarification."*

We appreciate the reviewer's insightful comment. The GCD curve in **Fig. 2a** is truncated because it starts only from the open circuit voltage (OCV = -0.16 V) and extends toward the "negative" voltage (under "open" polarization) to -1 V. In the tested system, the electrodes are made of different materials, each characterized by a distinct electrochemical potential (in contrast to symmetrical EDLCs, where $OCV \approx 0$ V). The OCV represents the natural equilibrium state of the system, and proper charging is considered to begin from this voltage. Starting the charging process from 0 V would require partially charging the system to this voltage (with reversed polarization), then discharging it to -0.16 V before recharging it toward -1 V. We also added appropriate clarification of scan rates, recalculations from the GCD technique in the revised SI and a reference to this part in the manuscript.

Please see the highlighted changes in line 200 of the Manuscript and lines 295-302 of the SI.

Comment 12: *“Line 249: The statement seems to refer to “the asymmetric cell as in Fig. 2a.”*

Thank you for your observation. We have added this information to the manuscript.

Please see the highlighted changes in line 233 of the Manuscript.

Comment 13: *“In Lines 276-284, the authors discuss the influence of electrode pore size on rectification ratios. This discussion should be expanded to include the effects of the pore size of the carbon electrode on redox reactions occurring at the Ti electrode in CAPodes.”*

We agree that the influence of the carbon electrode's pore size on redox reactions at the Ti electrode in CAPodes deserves further discussion. We have expanded the relevant section in the manuscript (**Lines 276-284**) to include an analysis of how the pore size of the carbon electrode can affect ion transport and, consequently, the redox reactions at the Ti electrode.

Please see the highlighted changes in lines 259-267 of the Manuscript.

Comment 14: *“As previously commented in the major revision, additional interpretation of Figs. 2g-2i is necessary, including any observations related to different electrolytes' performance.”*

Thank you for reiterating the need for further interpretation of **Fig. 2g-i**. We apologize for not addressing this adequately in the previous version of the manuscript. In response, we have expanded our discussion of **Fig. 2g-i** to include a detailed interpretation of the data

presented. Also, we have added the chapter “PWA as an optimized electrolyte” to the SI for further details.

Please see the highlighted changes in lines 276-288 of the Manuscript and lines 400-455 of the SI.

Comment 15: *“Additional experimental details are needed for in-situ Raman tests presented in Figs. 3a-3c.”*

We have added a paragraph on the in-situ measurements containing this information in the Experimental Section to provide clarity on the procedures used for **Fig. 3a-c**.

Please see the highlighted changes in lines 77-91 of the SI.

Comment 16: *“The discussion of rectification mechanisms in Fig. 3 is well-supported by advanced in-situ techniques, yet the study lacks innovative insights into the redox reaction mechanism. The author should focus more on the asymmetric mechanisms in CAPodes, such as further exploring the role of the C electrode in triggering Ti-side redox reactions.”*

The controllable orientation of unidirectional capacity in CAPode is achieved through asymmetric polarization between a plane metal and a porous carbon electrode, enabling selective redox reaction on the metal electrode surface. The polarization of the carbon electrode highly depends on SSA of carbon in the experiment, which then adjusts the potential of Ti to go far or near the redox potential of W^{6+}/W^{5+} . Furthermore, adjusting the mass loading or SSA of carbon electrodes can also regulate the EDL, which can balance charges from the faradaic redox process of W^{6+}/W^{5+} , optimizing the charge storage. In the revised manuscript, we added a discussion about the effects of the carbon electrode's pore size on redox reactions of electrolyte ions occurring on the metal plane.

Please see the highlighted changes in lines 259-267 of the Manuscript.

Comment 17: *“In line 402, the authors mention using Ti powder for printed electrodes. Clarification is need on how this material differs from the other Ti materials discussed.”*

In response, we have explained the differences between the Ti powder used for printed electrodes and the other Ti materials discussed in the section “1. Materials” in SI.

Please see the highlighted changes in lines 68-75 of the SI.

Comment 18: *“Including an actual image of the ipCAPode or NAND logic circuit setup would enhance the paper's impact.”*

We agree that including an actual image of the ipCAPode or the NAND logic circuit setup would provide visual context and enhance the manuscript's impact. In response, we have added a photograph of the experimental setup in the revised SI as **Fig. S21** and **S28**, specifically showing the layout and configuration of the ipCAPode and ipG-Cap.

Please see the highlighted changes in lines 614-616 and 737-739 of the SI.

Comment 19: *“What is the impact of the electrochemical window on NAND logic circuits or ipG-Cap architecture?”*

The electrochemical window is a critical parameter in the performance and functionality of NAND logic circuits or ipG-Cap architecture. This parameter influences various aspects of ion transistor behavior, including charge carrier control, threshold voltage, and ion sensitivity. The electrochemical potential directly influences the redox states of W ions, and the conductance peaks at specific redox potentials.¹⁻² For logic circuits like ipG-Cap, maintaining a clear distinction between the “on” and “off” states requires well-defined switching behavior. A stable electrochemical window **enhances the predictability of this switching**, leading to more reliable logic outputs. Operating within an optimal electrochemical window minimizes energy loss due to side reactions, thereby **improving the energy efficiency** of the ipG-Cap architecture. This efficiency is crucial for scaling the architecture in practical applications, as it directly impacts power consumption. **The**

threshold voltage of ipG-Cap can be precisely tuned by chemically controlling the electrochemical potential at the GE. This tunability is crucial for preparing further logic gates, like inverters and amplifiers with improved performance metrics.³⁻⁴

We have added a sentence on this in the manuscript.

Please see the highlighted changes in lines 417-420 of the Manuscript.

Comment 20: “*Could you provide the cycle stability of CAPodes in this structure as NAND logic circuits or in an ipG-Cap architecture?*”

We recognize the importance of demonstrating these structures’ durability and performance consistency over extended cycling, particularly for practical applications. In response, we have conducted cycle stability tests for CAPode with 1 M PWA as an electrolyte, and have provided detailed results in the revised manuscript. We measured key performance metrics such as the on/off current ratio at the voltage of ± 1 V (transconductance 1000 cycles) for CAPode in the G-Cap, which keeps more than 84% of RR_I after 1000 cycles. The new data are presented in Fig. S25, a short description is also added to the manuscript and SI.

Please see the highlighted changes in lines 318-320 and 349-350 of the Manuscript and lines 642-656 of the SI.

Comment 21: “*It is recommended that the authors discuss limitations on practical applications of the electrochemical capacitor diode and explore its potential future developments.*”

We agree that acknowledging the limitations and exploring potential future developments will provide a more comprehensive perspective on the electrochemical capacitor diode.

Limitations on practical applications

A semiconductor diode (e.g. 4H-SiC p-i-n) of 1 mm² conducted at 0.5 A with a current pulse FWHM of about 180 ns, a full width of about 300 ns, a rise time of about 10 ns, and

a fall time of about 200 ns has quite short response time and works extremely fastly.⁵ However, chemical circuits with iontronic devices offer significant benefits in various applications, including drug delivery (with timescales ranging from minutes to days) neuromodulation (seconds to hours), controlled polymerization (minutes to hours), intracellular calcium oscillations (microseconds to days), and plant development (minutes to days). Therefore, electrochemical capacitor diode (CAPode)-based logic circuits are a sort of ionic circuits, that utilize operated ions for signal processing instead of electrons, also have this limitation (low speed) on practical applications, which was discussed in the introduction part of the manuscript.

Potential future developments

Neuromodulation, as an important application of ion diodes/transistors, has attracted considerable attention in industry and academia. An appealing feature of neuromorphic devices based on electrochemical transistors is their potential for low power consumption per switching event.⁶ This was demonstrated in devices based on polymer nanofibres that exhibit energies as low as 1.23 fJ per synaptic spike.⁷ However, great efforts have been made to develop traditional electronic devices, but they still consume orders of magnitude more energy (generally above picojoule level) than do natural synapses.⁸ As a result, comparable electrochemical ionologic devices, such as CAPodes and G-Cap require less energy than conventional electronic devices as technology advances.

For scalability, various printed ionic circuits have recently been proposed, such as printed soft and pliable ionic circuit boards,⁹ stretchable ultraviolet curable ionic conductive elastomers for 3D printing approach to fabricate complex 3D flexible electronics,¹⁰ a 3D ionic microgel printing for manufacturing various ionic units¹¹ and so on. Furthermore, two different CAPode or G-Cap devices in the micrometer size based on EDLCs were manufactured using advanced 3D printing techniques.^{12,13} Future studies could explore advanced materials with higher rectification ratios and better stability, potentially enhancing the device's performance and operational life. For achieving higher rectification ratios, the development of new structural designs, such as hybrid architectures combining ECs and battery elements, could expand the range of applications. Because the response

time depends on the mobility of these ion species, different electrolyte systems should be explored, including ion size, solubility, viscosity, temperature, and conductivity.

We have added a paragraph on this to the manuscript.

Please see the highlighted changes in lines 449-455 of the Manuscript.

Reviewer #2:

“This paper introduces an innovative asymmetric electrochemical charge storage system, termed CAPode, which selectively interacts with redox-active ions to achieve unidirectional charge storage and exhibits rectifying characteristics. By controlling different redox couples, the operating voltage range can be adjusted. The selection of 1M phosphotungstic acid as the optimized electrolyte demonstrates high rectification ratios and low costs. Experimental validation shows the potential application of CAPode in the integration of logic gates, highlighting the promise of ion-logic devices in integrated circuits. The paper is clearly written with rigorous logic, making a significant contribution to the fields of electrochemical charge storage and logic operations, and exhibiting a high level of innovation and academic value. Hence, I recommend the manuscript to be published with major revisions addressed. The comments are as follows which would improve the quality of the manuscript:”

Thank you for your positive and constructive evaluation of our manuscript. We appreciate your acknowledgment of the CAPode’s innovative design, its selective interaction with redox-active ions, and its rectifying characteristics, as well as the potential of our approach for integration into ion-logic devices within circuit applications.

We agree that addressing the additional comments you have provided will further strengthen the manuscript. We have undertaken a series of major revisions in response to your feedback, including further elaboration on the rationale behind the choice of phosphotungstic acid as the optimized electrolyte and additional experimental data supporting the rectification ratios and operating voltage range adjustments. These

enhancements will clarify the logic and innovation of our approach in both electrochemical charge storage and logic operations.

Thank you once again for your valuable insights and suggestions, and we hope that the revised manuscript meets your expectations for publication.

Comment 1: *“The authors introduced ionic logic devices in the introduction, it is essential for the author to introduce the preliminary concept of ion electronics. Relevant articles to reference include: National Science Review 2024, nwa322; Materials Today 2024, 74, 187-202.”*

We appreciate the recommendation to introduce the preliminary concept of ion electronics in the introduction to enhance the context for our work on ionic logic devices. We have now incorporated a reference to the relevant articles you suggested that provides a foundational overview of ion electronics *National Science Review 2024, nwa322* and *Materials Today 2024, 74, 187-202*.

Please see the highlighted reference in line 63 and 593-597 of the Manuscript.

Comment 2: *“An electrolyte 1 M phosphotungstic acid (H₃PW₁₂O₄₀; PWA) with Keggin structure, for this electrolyte, the reason why this electrolyte was chosen and what are the advantages of this electrolyte for constructing supercapacitor diodes. This electrolyte is not widely available, what is his cost and it is necessary for the authors to explain this.”*

We selected 1 M phosphotungstic acid (H₃PW₁₂O₄₀; PWA) due to its unique Keggin structure, which enables reversible redox-active behavior, crucial for achieving high rectification ratios and unidirectional charge storage in our CAPode device. Additionally, the high proton conductivity and electrochemical stability of PWA enhance the overall performance and reliability of the supercapacitor diode. We acknowledge that PWA may not be as widely available as other electrolytes. However, its cost and toxicity are relatively low for laboratory-scale applications, making it a feasible choice within a research setting. In the revised manuscript, we have added a short explanation in the “2.1. System

construction” section and a detailed explanation “PWA as an optimized electrolyte” to the SI, addressing these points to clarify the rationale and advantages of this electrolyte choice.

Please see the highlighted changes in lines 124-129 of the Manuscript and lines 400-455 of the SI.

Comment 3: “As shown in Figure 2i, the rectification ratio of the supercapacitor diode based on Zn(TFSI)₂ is also quite high. The authors could present the performance of this device.”

Thank you for your suggestion to provide additional performance data for Zn(TFSI)₂-based CAPode that would offer valuable insights into the device’s capabilities. In response, we have added a detailed discussion of the performance metrics for the Zn(TFSI)₂-based device, including its rectification ratio and relevant electrochemical properties. This information is now included in the “2.2 *Electrochemical performance*” section, alongside **Fig. 2i**, to give a more comprehensive understanding of its performance. More details are also given in the new chapter “PWA as optimized electrolyte” in the SI.

Please see the highlighted changes in lines 276-281 of the Manuscript and lines 400-455 of the SI.

Comment 4: “As shown in Figure 3b, when the voltage drops to -1V, the Raman peak intensity significantly decreases, and the authors should explain this phenomenon. Additionally, is it necessary to perform normalization? Furthermore, the authors did not present the changes in the Raman peaks under positive voltage conditions.”

To meet the requirement of the journal (because of limits in the number of figures and number of words), we shifted **Fig. 3** into SI (now as **Fig. S16**). We have added an explanation in the revised manuscript discussing the observed decrease in Raman peak intensity at -1 V. This reduction likely results from reduced ion concentration or structural

changes within the electrode material under negative voltage, affecting the Raman-active modes. We agree that normalization could help enhancing clarity and have now applied it to the Raman spectra to better illustrate relative changes in intensity. Additionally, we appreciate your suggestion to examine Raman peak changes under positive voltage conditions. However, there is no redox activity in the positive voltage range, and its Raman spectrum in the positive range would be similar to 0 V. Therefore, our work presented the Raman spectrum at 0 V, which can be representative here. We have now included this analysis in **Fig. S16b** and provided a description in the “2.2 *Electrochemical performance*” section, offering a more balanced comparison of the material’s response under both polarities.

Please see the highlighted changes in lines 77-91 and lines 491-493 of the SI.

Comment 5: *“Notice that the authors only show performance up to 100 mV s⁻¹ scan rate. In fact, one of the most critical factors for ion devices is the rate of ion motion, and it is necessary for the authors to show CV curves for higher scan rates.”*

We appreciate the reviewer’s insight regarding the importance of ion motion rates for ion devices. However, we chose not to include scan rates above 100 mV s⁻¹ because at higher scan rates, the rectification ratio I (RR_i) will take on a value that does not meet the requirements for CAPodes (**Fig. S9a**; blue line), and the electrochemical response (CV curve in **Fig. S6c**) no longer resembles the characteristic behavior of the CAPode system. At 100 mV s⁻¹, a high current is observed even at 0 V (under system repolarization from -1 to 1 V), and the redox reaction extends up to 0.8 V, maintaining the distinct CAPode response. Therefore, the CAPodes based on faradic reactions can achieve high RR when the scan rate is not so high. This fact is also related to the dimensions of the device which is neither miniaturized nor optimized with respect to rate performance. Data at higher scan rates would not contribute in a meaningful way to the analysis of the CAPode’s unique performance characteristics. As we mentioned in the introduction, the operation rate of CAPode or G-Cap is moderate, not comparable with the semiconductors diode or transistors, but they have potential on the application of neuromodulation (seconds to hours). Due to the high capacitance, they could also be beneficial in power management.

Most nerve systems operate via ion-flux and downscaling enhances the signal transport rate in axons up to 100 m/s. The field of iontronics is still at an early stage and more applications may emerge in the near future.

Comment 6: *“The series of logic gates developed by the authors represents a significant breakthrough in the application of supercapacitor diodes. The authors could explore the advantages of ion-integrated circuits based on these constructions, as well as potential future application scenarios. For example, energy storage chips. Relevant articles to reference include: *Materials Today* 2024, 74, 187-202.”*

In response, we have expanded the discussion section to highlight the benefits of ion-integrated circuits, such as enhanced energy efficiency, compact design, and compatibility with bio-inspired and flexible electronics. We also address potential applications, including energy storage chips and neuromodulation with required speed from seconds to hours, as you suggested. Additionally, we have cited the recommended article (*Materials Today* 2024, 74, 187-202) to support these points.

Please see the highlighted changes in lines 449-455 of the Manuscript.

Reviewer #3:

“The authors present the CAPode, an asymmetric electrochemical charge storage device that uses selective redox reactions on a Ti electrode for one-way charge storage. This system is adjustable within a specific voltage range and exhibits rectification characteristics, which make it highly promising for integration into logic gates and potential capacitive computing applications with ion-selective redox devices.

Overall, I find it an interesting and creative approach to use redox chemistry and asymmetric electrode geometry to achieve current rectification. The concept is well-presented, though I believe the manuscript could be strengthened if the authors more clearly distinguished the novelty of this work compared to their previous reports and recent literature on redox gating and organic electrochemical transistors. It would be helpful to

highlight what aspects are unique in this study that advance the field beyond existing literature. For instance, they might consider clarifying the differences from key prior works such as Lochmann et al., 2020 (*Adv. Funct. Mater.*), Gellrich et al., 2024 (*Adv. Mater.*), and others listed.

- 1.S. Lochmann, Y. Bräuniger, V. Gottsmann, L. Galle, J. Grothe, S. Kaskel, Switchable Supercapacitors with Transistor-Like Gating Characteristics (G-Cap). *Adv. Funct. Mater.* 2020, 30, 1910439. <https://doi.org/10.1002/adfm.201910439>
- 2.C. Gellrich, L. Shupletsov, P. Galek, A. Bahrawy, J. Grothe, S. Kaskel, A Precursor-Derived Ultramicroporous Carbon for Printing Iontronic Logic Gates and Super-Varactors. *Adv. Mater.* 2024, 36, 2401336. <https://doi.org/10.1002/adma.202401336>
- 3.Bahrawy, P. Galek, C. Gellrich, J. Grothe, S. Kaskel, Advanced Redox Electrochemical Capacitor Diode (CAPode) Based on Parkerite ($\text{Ni}_3\text{Bi}_2\text{S}_2$) with High Rectification Ratio for Iontronic Applications. *Adv. Funct. Mater.* 2024, 34, 2405640. <https://doi.org/10.1002/adfm.202405640>
- 4.E. Zhang, N. Fulik, G.-P. Hao, H.-Y. Zhang, K. Kaneko, L. Borchardt, E. Brunner, S. Kaskel, *Angew. Chem. Int. Ed.* 2019, 58, 13060.
- 5.P. Tang, W. Tan, F. Li, S. Xue, Y. Ma, P. Jing, Y. Liu, J. Zhu, X. Yan, A Pseudocapacitor Diode Based on Ion-Selective Surface Redox Effect. *Adv. Mater.* 2023, 35, 2209186. <https://doi.org/10.1002/adma.202209186>
- 6.Ma, H., Liang, J., Qiu, J., Jiang, L., Ma, L., Sheng, H., Shao, M., Wang, Q., Li, F., Fu, Y., Wang, J., Xie, E., Chai, Y. and Lan, W. (2023), A Biocompatible Supercapacitor Diode with Enhanced Rectification Capability toward Ion/Electron-Coupling Logic Operations. *Adv. Mater.*, 35: 2301218. <https://doi.org/10.1002/adma.202301218>
- 7.L. Zhang, C. Liu, H. Cao, A. J. Erwin, D. D. Fong, A. Bhattacharya, L. Yu, L. Stan, C. Zou, M. V. Tirrell, H. Zhou, W. Chen, Redox Gating for Colossal Carrier Modulation and Unique Phase Control. *Adv. Mater.* 2024, 36, 2308871. <https://doi.org/10.1002/adma.202308871>
- 8.Hui Cao, Changjiang Liu, Dillon D. Fong, Anand Bhattacharya, Matthew V. Tirrell, Hua Zhou, Wei Chen; Redox gating-induced modulation of charge carrier density and lattice expansion in LaNiO_3 thin films. *Appl. Phys. Lett.* 12 August 2024; 125 (7): 074101. <https://doi.org/10.1063/5.0217899>

- 9. Ahsan Raza, Umar Farooq, Khalida Naseem, Sarfaraz Alam, Mohammad Ehtisham Khan, Akbar Mohammad, Waleed Zakri, Muhammad Yasir Khan, (2024), A focused review on organic electrochemical transistors: A potential futuristic technological application in microelectronics, *Microchemical Journal*, 207, 111737. <https://doi.org/10.1016/j.microc.2024.111737>.
- 10. Rivnay, J., Inal, S., Salleo, A. et al. Organic electrochemical transistors. *Nat Rev Mater* 3, 17086 (2018). <https://doi.org/10.1038/natrevmats.2017.86>
- 11. Guo, J., Chen, S.E., Giridharagopal, R. et al. Understanding asymmetric switching times in accumulation mode organic electrochemical transistors. *Nat. Mater.* 23, 656–663 (2024). <https://doi.org/10.1038/s41563-024-01875-3>
- 12. Huang, W., Chen, J., Yao, Y. et al. Vertical organic electrochemical transistors for complementary circuits. *Nature* 613, 496–502 (2023). <https://doi.org/10.1038/s41586-022-05592-2>

Thank you for your thoughtful feedback and for highlighting the potential of our CAPode device. We appreciate your suggestion to more clearly distinguish our work's novelty from previous studies and recent literature on redox gating and organic electrochemical transistors.

In response, we have revised the introduction and discussion sections to explicitly clarify how our approach builds upon and diverges from key prior works. Specifically, we emphasize the unique aspects of our study **asymmetric electrode design** and **selective redox reaction**. Unlike previous studies on CAPodes, our work exploits the asymmetry of the system to achieve the redox potential. This configuration enhances unidirectional charge storage and enables tunable rectification within a specified voltage range. By utilizing phosphotungstic acid ($H_3PW_{12}O_{40}$; PWA) as the electrolyte, we achieve rectification based on ion-selective redox behavior, a feature that is distinct from traditional organic electrochemical transistors and recent redox-gating studies.

We are grateful for your detailed feedback, which has enabled us to strengthen our manuscript by clearly positioning our study within the current body of research.

A detailed comparison has been included in the section “comparing systems” in the SI (section 16).

Comment 1: *“The technical characterizations presented, such as electrical measurements and X-ray/spectroscopy studies, are well done. However, it is surprising that the manuscript doesn’t include a single real photograph of the device, nor any SEM or morphological images. For instance, Figures 1, 4a, and S24 are helpful schematics, but readers would benefit from seeing actual images of the device, as this might also help explain aspects like the “high inner resistance of printed cell compared to coin cell” mentioned on Line 512. Including images of printed patterns could add clarity.”*

In response, we have added photographs images of the device and printed patterns to the supplementary information (section 15; **Fig. S21** and **S28**). These images provide a clearer view of the device structure and surface morphology, offering insight into the printed patterns that contribute to the observed differences in internal resistance. We believe these additions will improve the readers' understanding of the device's unique characteristics.

Please see the highlighted changes in line 301 and 386 of the Manuscript and lines 614-616 and 737-739 of the SI.

Comment 2: *“There are a few areas where the figure quality could be improved. For example, Figure S1 is challenging to interpret due to low inset figure readability. Improving resolution or font size in these areas would enhance readability. Additionally, a few minor textual improvements are needed, such as moving the abbreviation “SSA” in Line 117 to the correct location.”*

We have improved the resolution and font size of Figure S1 to enhance readability, especially in the inset areas. Additionally, we have corrected the placement of the abbreviation “SSA” on Line 117 and reviewed the manuscript for other minor textual adjustments to ensure accuracy and clarity.

Please see the highlighted changes in lines 105-106 of the Manuscript and line 115 of the SI.

Comment 3: *“It might also be helpful to revise the structure of some sections for better clarity. For example, on Line 187, discussions around SI content could be streamlined by either moving the equations and parameters to the SI entirely or bringing the related discussion into the main text, as appropriate. “Rectification ratio” is referenced frequently throughout but lacks a clear definition in the main text.”*

We agree that revising the structure will enhance clarity. Following your suggestion, we have streamlined discussions around rectification ratios. Specifically, we have moved the discussion fully to the Supporting Information (SI) or incorporated the necessary context into the main text where it directly contributes to the narrative flow. Additionally, we have included a clear definition of the “rectification ratio” early in the main text to provide a solid foundation for its frequent references throughout the manuscript.

Please see the highlighted changes in lines 178-183 of the Manuscript and lines 337-340 of the SI.

Comment 4: *“In Figure 3f, the XANES data for tungsten could benefit from additional data points below the L3 absorption band. Recording only ~50 eV below the band may limit the modeling accuracy and affect the baseline adjustment. Also, in the SI, there’s a lack of detail on data processing for this XANES data—providing specifics would add transparency.”*

To address the reviewer’s comment, we appreciate the suggestion regarding additional data points below the L3 absorption edge for tungsten. We recognize that recording a broader range below the edge could potentially improve the baseline modeling and overall accuracy. However, our primary focus was on the near-edge region to capture the oxidation states effectively, and this range was selected based on initial assessments of

signal relevance for the redox processes we are studying. Expanding the range would indeed offer finer detail, which is presented in **Fig. S20**.

Regarding data processing, we have now included further specifics in the SI (see description on Fig S16f and S17), detailing the methods used for baseline adjustment, normalization, and peak fitting. This additional information should provide the clarity and transparency requested for the processing of XANES data.

Please see the highlighted changes in lines 467, 510, and 566-573 of the SI.

Comment 5: *“Line 359: The authors refer to the W K3 edge, which I believe should be the L3 edge, as the W K3 edge is around 69.5 keV.”*

Thank you for identifying this shortcoming. You are correct, the reference should indeed be to the tungsten L3 edge rather than the K3 edge. We have corrected this in the revised manuscript to avoid any confusion.

Please see the highlighted changes in lines 467 and 510 of the SI.

Comment 6: *“Line 383: This section could benefit from more interpretation. What is the main conclusion, or purpose, of this experiment?”*

We have revised this section to better highlight the primary purpose of this experiment: to demonstrate the application of CAPode devices in logic gate circuits. The experiment underscores how the rectification behavior of CAPodes can be harnessed to perform basic logic functions, thereby showcasing their potential in developing ion-based integrated circuits. We have added a paragraph to the manuscript and in the SI (description for **S27**) Adding these interpretive details will clarify the significance of the findings for readers.

Please see the highlighted changes in lines 359-364 and 382-383 of the Manuscript and lines 721-735 of the SI.

Comment 7: *“Line 405: When discussing the device’s time response, quantifying or fitting a time constant and comparing it to ion diffusion values could clarify how ion dynamics impact performance.”*

Thank you for your comment. We have evaluated time constants and included a discussion on this aspect in the SI (“Time Constant from the Transient Response Method”).

Please see the highlighted changes in lines 658-715 of the SI.

Comment 8: *“Line 423: Why does the ipCAPode show a more pronounced current response than the coin cell?”*

The more pronounced current response observed in the ipCAPode compared to the coin cell can be attributed to the enhanced ion accessibility and reduced path length within the printed architecture. The ipCAPode’s design allows for more efficient ion transport and interaction at the electrode interfaces, which results in a higher current response. In the revised manuscript, we have clarified this distinction and explain how the structural and transport characteristics of the ipCAPode contribute to this performance difference.

Please see the highlighted changes in lines 329-337 of the Manuscript.

Comment 9: *“In Figure 5, it is a bit unclear if the gate electrode (GE) is modulating the CE-WE current by changing the ion density or if another effect might be responsible, such as a transverse electric field or larger surface area effects. Additionally, the schematic in Fig. 5a does not match Fig. 5d, unless the GE is indeed placed on top of the channel. More clarity here would improve reader comprehension.”*

In response, the modulation of the CE-WE current by the additional external gate electrode (GE) is achieved by altering the local ion density rather than through a transverse electric field effect or surface area differences. In the G-Cap system, GE reversibly depletes/injects electrolyte ions into the working capacitor (W-Cap) channel,

effectively controlling its charge storage capacity, and also potentials of W-Cap electrodes. This ion density modulation directly impacts the ion distribution between the WE and CE, which is the primary mechanism for current control in the system. In the improved manuscript, the **Fig. 5** was changed to **Fig. 4** for meeting the length requirement of journal. We have swapped the order of **Fig. 4a** with **4d** (currently **Fig. 4a** and **4b**) and added a much more detailed explanation for the G-Cap mechanism. In the same paragraph, we also refer to reference no. 23 (Lochmann S., Bräuniger Y., Gottsmann V., Galle L., Grothe J., Kaskel S., "Switchable Supercapacitors with Transistor-Like Gating Characteristics (G-Cap)," *Advanced Functional Materials*, 09 March 2020, <https://doi.org/10.1002/adfm.201910439>), where readers can find more details about the G-Cap system.

Regarding the schematic discrepancy between **Fig. 4a** and **4b**, **Fig. 4a** is intended to present the mechanism of CAPode performance schematically. The placement of a drawing of GE above both W-Cap electrodes is meant to illustrate better the reversible depletion/injection of ions from the WE and CE inter-electrode space. Positioning GE next to (on the right side of) both W-Cap electrodes, as in **Fig. 4b**, would not clearly depict this phenomenon of ion depletion from the W-Cap inter-electrode space. **Fig. 4b**, in contrast, presents a scheme of the device that more closely reflects its actual structure, aiming to illustrate our proposed *ip*G-Cap configuration with separated electrodes that belong to the W-Cap half-system. In addition, this figure indicates the materials used for each electrode.

In summary, both illustrations (**Fig. 4a** and **4b**) serve distinct purposes to best achieve the intended explanation. We have added an explanatory paragraph and changed the figure caption.

Please see the highlighted changes in lines 386-404 of the Manuscript.

"In general, I recommend reconsideration after a major revision to address these points. Thank you for the opportunity to review this work, and I look forward to seeing how it develops with these refinements"

Thank you for the valuable suggestions. We hope the revision will meet your expectations.

Reviewer #1 (Remarks to the Author):

“Thank you very much for your response to my previous comments. The manuscript has improved significantly. The authors have addressed my comments point by point, and the additional data and further analysis provide strong support for the experimental results. I only have 2 minor points as follows.”

Dear Reviewer #1,

we sincerely thank you for your positive review and thoughtful comments on our manuscript.

Comment 1: *“As mentioned in the abstract, the most innovative aspect of this study lies in the fact that “this unidirectional capacity is achieved through asymmetric polarization between a plane metal and a porous carbon electrode, enabling selective redox reactions on the metal surface.” However, considering the current title of the manuscript, should it be more specific, such as “Redox-active Keggin-type electrolyte” be noted in abstract, or would “Redox-active electrolyte based printed ionologic devices” be a more suitable option as title?”*

Thank you for your thoughtful suggestion about the manuscript title and abstract. We really appreciate your attention to the key aspects of our study.

To keep a good balance between being specific and maintaining broader relevance, we’ve decided to change the title to “Redox-active electrolyte based printed ionologic devices” to highlight that various redox-active electrolytes can work in our system based on asymmetric polarization between a plane metal and a porous carbon electrode. We also incorporated “Redox-active Keggin-type electrolyte” in the abstract can improve clarity, as you suggested.

Please see the highlighted changes in the title of Manuscript and SI, and the line 31 of Manuscript.

Comment 2: “Regarding Comment 20, the device presented in this manuscript does not appear to exhibit good cycling stability, especially when compared to the similar work previously published by the authors' research group (*Advanced Functional Materials* 34.45 (2024): 2405640). For the device in this study, what are the primary factors limiting its cycling stability? Could the authors provide further discussion on this aspect?”

Thank you for your insightful question regarding the cycling stability of our device. We have to say that the stability observed in this study is comparable with our previous work (*Advanced Functional Materials* 34.45 (2024): 2405640), and we appreciate the opportunity to further discuss the limiting factors.

Fig. 1. Rectification ratio for systems with different switching sequences depending on the cycle number (*Advanced Functional Materials* 34.45 (2024): 2405640).

In Fig. 1, we would first like to clarify that different assessment methods were used to evaluate the stability in our previous study (*Advanced Functional Materials* 34.45 (2024): 2405640).

The first assessment method, based on cyclic voltammetry (CV) repolarization (blue line), evaluates rectification through repeated voltage sweeps. In this approach, the potential is continuously swept over a range for several minutes, allowing the electrode-electrolyte interface to gradually stabilize.

The second assessment method focuses on transconductance cycling by using chronoamperometry (CA) measurement (red and green line), where the on/off current ratio is monitored under a fixed voltage condition. The transconductance cycling method fixes the voltage at ± 1 V only for 10 s.

In this study, we used the second method. Therefore, we are going to compare the stability performance of two works in the same CA assessment method.

To ensure a fair comparison of stability performance, we will focus on results obtained using the same CA assessment method. The stability data from our previous work is represented by the red and green lines in Fig. 1. For instance, the red line shows that after 100 cycles, the rectification ratio remained stable at 100% (9.6). However, after 1000 and even 5000 cycles, the rectification ratio decreased to approximately 70%, which is comparable to the results in this study (around 80%).

The primary factor limiting cycling stability may stem from the cell setup. In this study, we used a parafilm-fixed for long-term cycling, which can lead to solvent evaporation. Since the mechanism described in this work involves the redox-active ions from the electrolyte, solvent retention plays a crucial role in maintaining system stability.

Other possible factors limiting cycling stability of this, which also have been reported for other electrochemical systems based on redox reactions of electrolytes, include side reactions such as hydrogen/oxygen evolution and the degradation of oxygen-sensitive redox species upon exposure to air (*Journal of Power Sources* 397, 2018, 214-222; *Journal of Energy Chemistry* 96, 2024, 89-109).

We have added the primary factor (above highlight part) in the SI below the Fig. S25.

Reviewer #2 (Remarks to the Author):

“I have read all the replies the author answered in the review report. The author addressed all my concerns in great detail, and the revised manuscript is suitable for the publication. I suggest the publication of this manuscript at the current version.”

Dear Reviewer #2,

thank you for your thoughtful review and for recognizing the efforts we put into revising the manuscript. We truly appreciate your valuable comments and suggestions, which have helped us improve the quality of our work.

We are pleased to hear that you find the revised manuscript suitable for publication. Your feedback has been instrumental in refining our study, and we sincerely appreciate your time and expertise.

Reviewer #3 (Remarks to the Author):

“Thank the authors for carefully considering my comments and addressing them thoughtfully. I appreciate the effort you have put into refining the manuscript. The revisions have significantly strengthened the clarity and impact of your work. I fully support the publication of this manuscript in Nature Communications.”

Dear Reviewer #3,

thank you for your kind and supportive feedback. We sincerely appreciate the time and effort you have taken to review our manuscript. Your thoughtful comments and suggestions have greatly contributed to improving the clarity and impact of our work.

We are grateful for your endorsement of our revised manuscript and your recommendation for publication in Nature Communications. Your insights have been invaluable, and we truly appreciate your support.